# Dynamic processes in the magnetic field and in the ionosphere during the 30 August–2 September, 2019 geospace storm: Influence on HF radio wave characteristics

Yiyang Luo[1], Leonid Chernogor[2], Kostiantyn Garmash[2], Qiang Guo[3], Victor Rozumenko[2], Yu Zheng[4]

[1]Department of Theoretical Radio Physics, V. N. Karazin Kharkiv National University, Kharkiv, 61022, Ukraine
[2]Department of Space Radio Physics, V. N. Karazin Kharkiv National University, Kharkiv, 61022, Ukraine
[3]Harbin Engineering University, 145 Nantong Street, Nangang District, Harbin, 150001, China
[4]Qingdao University, 308 Ningxia Road, Qingdao, 266071, China

*Correspondence to*: Yu Zheng (zhengyu@qdu.edu.cn)

**Abstract.** The concept that geospace storms are comprised of synergistically coupled magnetic storms, ionospheric storms, atmospheric storms, and storms in the electric field originating in the magnetosphere, the ionosphere and the atmosphere (i.e., electrical storms) was validated a few decades ago. Geospace storm studies require the employment of multiple-method approach to the Sun–interplanetary medium–magnetosphere–ionosphere–atmosphere–Earth system. This study provides general analysis of the 30 August–2 September 2019 geospace storm, the analysis of disturbances in the geomagnetic field and in the ionosphere, as well as the influence of the ionospheric storm on the characteristics of HF radio waves over the People's Republic of China. The main results of the study are as follows. The energy and power of the geospace storm have been estimated to be $1.5 \times 10^{15}$ J and $1.5 \times 10^{10}$ W, and thus this storm is weak. The energy and power of the magnetic storm have been estimated to be $1.5 \times 10^{15}$ J and $9 \times 10^{9}$ W, i.e., this storm is moderate, and a characteristic feature of this storm is the duration of the main phase, of up to two days. The recovery phase also was lengthy, no less than two days. On 31 August 2019 and on 1 September 20197, the variations in the $H$ and $D$ components attained 60–70 nT, while the $Z$-component variations did not exceed 20 nT. On 31 August 2019 and on 1 September 2019, the level of fluctuations in the geomagnetic field in the 100–1000 s period range increased from 0.2–0.3 nT to 2–4 nT, while the energy of the oscillations showed a maximum in the 300–400 s to 700–900 s period range. During the geospace storm, a moderate to strong negative ionospheric storm was manifested itself by the reduction in the ionospheric $F$ region electron density on 31 August 2019 and 1 September 2019 by a factor of 1.4 to 2.4 times as compared to the its values on the reference day. Appreciable disturbances were also observed to occur in the ionospheric $E$ region, and possibly in the $E_s$ layer. In the course of the ionospheric storm, the altitude of reflection of radiowaves could sharply increase from ~150 km to ~300–310 km. The atmospheric gravity waves generated within the geospace storm modulated the ionospheric electron density; for the ~30 min period oscillation, the amplitude of the electron density disturbances could attain ~40%, while it did not exceed 6 % for the ~15 min period. At the same time, the height of reflection of the radio waves varied quasi-periodically with a 20–30-km amplitude. The results obtained have made a contribution to understanding of the geospace storm physics, to developing theoretical and empirical models of geospace storms, to the acquisition of detailed understanding of the adverse effects that geospace storms have on radiowave propagation and to applying that knowledge to effective forecasting these adverse influences.

## 1 Introduction

Geospace storms are comprised of synergistically coupled magnetic storms, ionospheric storms, atmospheric storms, and storms in the electric fields originating in the magnetosphere, the ionosphere, and the atmosphere (i.e., electrical storms) (Chernogor and Rozumenko, 2008; Chernogor, 2011; Chernogor and Domnin, 2014). Consequently, the discussion of only one of the storms would be incomplete, and therefore, the analysis of geospace storms requires the employment of a systems approach. These storms are of solar origin, and they may be accompanied by solar flares, coronal mass ejections, high speed

solar wind streams, energetic proton fluxes, and solar radio bursts. All listed above processes affect the magnetosphere, the ionosphere, the atmosphere, and the internal terrestrial layers through the interplanetary medium. Their joint study requires clustered-instrument studies of the internal layers in the Sun–interplanetary-medium–magnetosphere–ionosphere–atmosphere–Earth (SIMMIAE) system (Chernogor and Rozumenko, 2008; Zalyubovsky et al., 2008; Chernogor, 2011; Chernogor and Domnin, 2014; Chernogor and Rozumenko, 2011, 2012, 2014, 2016, 2018; Chernogor et al., 2020). The study of geospace storms, which are not quite correctly termed by some authors as the magnetic storms, the ionospheric storms, or thermospheric storms, has almost a 100 year history. The proper magnetic storms have been observed for about 400 years. The results of the first observations of ionospheric disturbances occurring during magnetic storms were described by Hafstad and Tuve (1929) and Appleton and Ingram (1935).

Matsushita (1959) was the first to apply statistics to ionospheric storms. Later, the statistical approach was employed by Chernogor and Domnin (2014). The statistics of magnetic and ionospheric storms is presented in (Vijaya Lekshmi et al., 2011; Yakovchouk et al., 2012; Zolotukhina et al., 2018).

A few authors (Danilov and Morozova, 1985; Prölss, 1995, 1997; Laštovička, 1996; Fuller-Rowell et al., 1997; Buonsanto, 1999; Danilov and Laštovička, 2001; Danilov, 2013) generalized the observations of ionospheric storms.

The results of recent studies of ionospheric storm effects are presented in a large number of papers (see, e.g., Blanch et al., 2005; Mendillo, 2006; Pirog et al., 2006; Prölss, 2006; Kamide and Maltsev, 2007; Borries et al., 2015; Liu et al., 2016; Polekh et al., 2017; Shpynev et al., 2018; Stepanov et al., 2018; Yamauchi et al., 2018; Blagoveshchensky and Sergeeva, 2019; Chernogor et al., 2020; Mosna et al., 2020).

In particular, the studies of the 7–8 September 2017 geospace storm are presented in the papers (Yamauchi et al., 2018; Blagoveshchensky and Sergeeva, 2019; Mosna et al., 2020; Habarulema et al., 2020).

Many authors have employed the systems approach to the SIMMIAE system over the last 40 years. The basics of the systems paradigm are stated and validated by Chernogor and Rozumenko [2008, 2011, 2012, 2014, 2016, 2018], Chernogor [2011], and Chernogor and Domnin [2014].

The study of geospace storms is of major scientific importance (Gonzalez et al., 1994; Knipp and Emery, 1998, Freeman, 2001; Space…, 2001; Benestad, 2002; Carlowicz and Lopez, 2002; Lathuillère et al., 2002; Feldstein et al., 2003; Bothmer and Daglis, 2006; Lilensten and Bornarel, 2006). Mechanisms for subsystem coupling, both positive and negative ones, in the SIMMIAE system, as well as feedback and precondition of the system components have not been sufficiently well studied. In particular, Gonzalez et al. (1994) made an excellent review summarizing information on geomagnetic storms up to the early 1990s. Since then, the understanding of geomagnetic storms has significantly advanced [Danilov, 2013]. The authors have used the relation given by Gonzalez et al. (1994) for the magnetic storm energy. Knipp and Emery (1998) described in detail the processes accompanying the November 2–11, 1993 geomagnetic storm. Feldstein et al. (2003) analyzed in detail the energy of the processes acting in the magnetosphere during two particular storms.

The dynamics of the processes, energy transfer, the appearance of trigger mechanisms for energy release, etc., remain not fully understood.

The study of geospace storms is also of special interest to estimate serious malfunctions in numerous systems: radar, telecommunications, radionavigation, radio astronomy, and in ground-based power system, etc. (Goodman, 2005). Storms have the potential to harm humans on the ground or in the near-Earth space environment. Modern society and human well-being become reliant more and more on space-based technologies, and consequently, on the state of space weather and geospace storms. The manifestations of geospace storms vary over the solar cycle, and depend on season, local time, latitude, longitude, and so on. Therefore, there is an urgent need to study each sufficiently large geospace storm. Such an investigation reveals both general storm properties and its specific features.

The purpose of this paper is to present a general analysis of the 30 August–2 September, 2019 geospace storm, to
analyze disturbances in the ionosphere and in the geomagnetic field, and to examine the influence of the ionospheric storm
on the characteristics of the HF radio wave propagating over the People's Republic of China area.
In this paper, a brief description of the instrumentation and the techniques employed is presented first. This is
followed by a general analysis of the space weather state, the magnetic and ionospheric storms. Next, a description of the
results of radio observations obtained at oblique incidence on the reference day and in the course of the geomagnetic storm is
examined in detail. Finally, the results of analysis of the geomagnetic storm features are discussed, and the main results are
listed.

## 89 2 Instrumentation and measurement techniques

### 90 2.1 Observational instruments

*Fluxmeter magnetometer.* The magnetometer is located at the Kharkiv V. N. Karazin National University Magnetometer
Observatory (49.64°N, 36.93°E). It acquires data on variations in the horizontal ($H$, $D$) geomagnetic field components in the
1–1000 s period range with a 0.5 s temporal resolution delivering 1 pT–1 nT sensitivity. The fluxmeter magnetometer is
described in detail by Chernogor (2014) and Chernogor and Domnin (2014).
*Three-Axis Fluxgate Magnetometer.* The LEMI-017 Meteomagnetic Station (49.93°N, 36.95°E) is located at the
Institute of Radio Astronomy of NASU Low Frequency Observatory (49.93°N, 36.95°E) [Magnetic field variations
http://geospace.com.ua/en/observatory/metmag.html, last access: 15 June 2020]. It takes measurements of the geomagnetic
field $H, D,$ and $Z$ components at 1 s interval with 10 pT sensitivity.
*Multi-frequency multipath system involving the software-defined radio for the oblique incidence radio sounding of*
*the ionosphere.* It is located at the Harbin Engineering University campus, the People's Republic of China (45.78°N,
126.68°E) (Chernogor et al., 2019a, b, c, 2020; Guo et al., 2019a, b, c, 2020; Luo et al., 2020a). The ionosphere is
continuously monitored over eleven radio paths utilizing emissions from broadcasting stations in the 5–10 MHz frequency
range and located in Japan, the Russian Federation, Mongolia, the Republic of Korea, and the People's Republic of China
(Fig. 1), the radio path lengths (Table 1) are found in the $(1–2) \times 10^3$ km distance range, and the signal reception and
processing is performed at the Harbin Engineering University.
*Ionosondes.* They are used to assess a general state of the ionosphere. The WK546 URSI code ionosonde at the City
Wakkanai (45.16°N, 141.75°E), Japan, is the closest to Harbin (Ionosonde Stations in
Japan: URL: wdc.nict.go.jp/IONO/HP2009/contents/Ionosonde_Map_E.html, last access: 15 June 2020). To assess the
characteristic extent of the ionospheric storm, the City of Moscow (the Russian Federtation) ionosonde data are used (List of
years for MOSCOW**:** https://lgdc.uml.edu/common/DIDBYearListForStation?ursiCode=MO155, last access: 15 June 2020).

### 111 2.2 Analysis techniques

The fluxmeter magnetometer data recorded initially on a relative scale have been converted into absolute values using the
magnetometer transfer function. Then, temporal dependencies of the geomagnetic field have been subjected to the systems

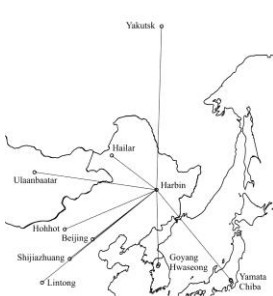

Figure 1: Layout of the propagation paths used for monitoring dynamic processes acting in the ionosphere.

spectral analysis, which employs simultaneously the short-time Fourier transform, the wavelet transform using the Morlet
wavelet as a basis function, and the Fourier transform in a sliding window with a width adjusted to be equal to a fixed number
of harmonic periods (Chernogor, 2008). Analysis of the obtained spectra follows.

Table 1

*Basic parameters of 11 radio paths used for probing the ionosphere at oblique incidence*. Retrieved from
https://fmscan.org/index.php

| Transmitter | | | | Propagation path midpoint | | |
|---|---|---|---|---|---|---|
| Frequency [kHz] | North latitude [deg.] | East longitude [deg.] | Location [country] | Distance to Harbin [km] | North latitude [deg.] | East longitude [deg.] |
| 5,000 | 34.95 | 109.56 | Lintong/ Pucheng (China) | 938 | 40.37 | 118.12 |
| 6,015 | 37.21 | 126.78 | Hwaseong (ROK) | 475 | 41.50 | 126.73 |
| 6,055 | 35.47 | 140.21 | Chiba/ Nagara (Japan) | 805 | 40.63 | 133.45 |
| 6,175 | 39.75 | 116.81 | Beijing (China) | 525 | 42.77 | 121.75 |
| 6,600 | 37.60 | 126.85 | Goyang (ROK) | 455 | 41.69 | 126.77 |
| 7,260 | 47.80 | 107.17 | Ulaanbaatar/ Khonkhor (Mongolia) | 748 | 46.79 | 116.93 |
| 7,345 | 62.24 | 129.81 | Yakutsk (Russia) | 923 | 54.01 | 128.25 |
| 9,500 | 38.47 | 114.13 | Shijiazhuang (China) | 655 | 42.13 | 120.41 |
| 9,520 | 40.72 | 111.55 | Hohhot (China) | 670 | 43.25 | 119.12 |
| 9,750 | 36.17. | 139.82 | Yamata (Japan) | 785 | 40.98 | 133.25 |
| 9,830 | 39.75 | 116.81 | Beijing (China) | 525 | 42.77 | 121.75 |

The Radio Astronomy of the National Academy of Sciences of Ukraine three-axis fluxgate magnetomer has been
used to control a general state of the geomagnetic field, and a specific signal processing procedure was not needed.
The data acquired by the multi-frequency multipath system for the oblique incidence radio sounding of the
ionosphere have been subjected to processing in detail, and the products included the universal time dependencies of the
Doppler spectra, the main ray amplitude, $A(t)$, and the Doppler shift of frequency, $f_D(t)$. Further, the $f_D(t)$ and $A(t)$ were
subjected to secondary processing to obtain the trends $\overline{f}_D(t)$ and $\overline{A}(t)$, the fluctuations $\delta f_D(t) = f_D(t) - \overline{f}_D(t)$,
$\delta A(t) = A(t) - \overline{A}(t)$, and the spectra in the period range $T \approx 1$–60 min and greater (Chernogor, 2008).

## 3 Analysis of the space weather state

The space weather variations under study are the event of CIR/CH HS origin combined with solar sector boundary crossing event, which could affect geomagnetic situation (see ftp://ftp.swpc.noaa.gov/pub/warehouse/2019/WeeklyPDF/prf2296.pdf; Koskinen, 2011). The data retrieved from https://omniweb.gsfc.nasa.gov/form/dx1.html have been used to analyze the solar wind parameters. On 29 August 2019, the proton density, $n_{sw}$, exhibited an increase from $\sim10^6$ m$^{-3}$ to $15 \times 10^6$ m$^{-3}$, and subsequently, a decrease from $15 \times 10^6$ m$^{-3}$ to $1\times10^6$ m$^{-3}$ in the course of the next three days (Fig. 2). In the course of 28 and 29 August 2019 and of the first half of 30 August 2019, the solar wind bulk speed, $V_{sw}$, varied from $\sim350$ km s$^{-1}$ to 500 km s$^{-1}$. After 12:00 UT on 30 August 2019 through about 01:00 UT on 1 September 2019, the $V_{sw}$ value exhibited an increase from $\sim400$ km s$^{-1}$ to 750 km s$^{-1}$ with a peak of 835 km/s observed early on 1 September 2019 (see ftp://ftp.swpc.noaa.gov/pub/warehouse/2019/WeeklyPDF/prf2296.pdf). During almost four days, $V_{sw} \approx 600$–750 km s$^{-1}$. Before 12:00 UT on 30 August 2019, the temperature, $T_{sw}$, of the solar wind particles was observed to be in the $(1$–$2) \times 10^5$ K range. After 12:00 UT on 30 August 2019, it showed an increase from $10^5$ K to $4.4 \times 10^5$ K in the course of 24 h, and eventually, fluctuating, it exhibited a gradual decrease from $4.4 \times 10^5$ K to $10^5$ K. As expected, the increases in $n_{sw}$ and $V_{sw}$ gave rise to an increase in the solar wind dynamic pressure, from $\sim0.2$ nPa to $\sim3$ nPa. The East–West $B_y$ and the North–South $B_z$ components of the interplanetary magnetic field exhibited fluctuations in the $-3$ nT to 8 nT and from $-7$ to 3 nT ranges, respectively. Since approximately 12:00 UT on 30 August 2019, the value of the $B_z$ component remained predominantly negative. This indicated that the magnetic storm ensued. Over the following day (from 08:00 UT on 30 August 2019 to 07:00 UT on 3 September 2019), energy input per unit time, $\varepsilon_A$, from the solar wind into the Earth's magnetosphere occasionally increased to 14–15 GJ s$^{-1}$; before the storm commencement, the $\varepsilon_A$ value did not exceeded 1 GJ s$^{-1}$.

The $K_p$ index values exhibited variations from 0 to 2 before the storm commencement, and from $\sim2$ to 5.7 over four days afterwards. Before the storm commencement, the $D_{st}$ index was observed to fluctuate in the $-10$ nT to 6 nT range. At about approximately 12:00 UT on 30 August 2019, $D_{st} \approx 12$ nT; from 10:00 UT to 14:00 UT, the storm commencement was observed to occur. After 20:00 UT on 30 August 2019, the $D_{st}$ values began to show a gradual decrease to $-55$ nT, which was attained at about 06:00 UT on 1 September 2019; over this time period, the storm main phase was observed to occur. After 06:00 UT on 1 September 2019, the storm transitioned to the recovery phase, which lasted for a few days. Thus, this magnetic storm was seen to be of quite a long duration over the last few years, but it was not the strongest, which is its main feature. A long duration ionospheric storm was expected to follow the longest duration magnetic storm. The geomagnetic and ionospheric storm features are described further in detail.

## 4 Analysis of the magnetic storm

### 4.1 Level of geomagnetic field variations

Magnetic measurements at the Institute of Radio Astronomy of NASU Low Frequency Observatory, Ukraine (49.93° N, 36.95° E) show that the state of the geomagnetic field was quiet on 29 August 2019 (panel (a) in Fig. 3). After 12:00 UT on 30 August 2019, relatively small, $\sim10$–20 nT, variations appeared in all geomagnetic field components (see panel (b) in Fig. 3). On 31 August 2019, the variations increased up to 60–70 nT (see panel (*c*) in Fig. 3). The Z component was changing less, no more than by 20 nT. The variations on 1 September 2019 remained approximately the same (see panel (d) in Fig. 3). The fluctuation excursions of the components significantly decreased on 2 September 2019 (see panel (e) in Fig. 3). In the course of the next two days, the magnetic field remained weakly disturbed (see panel (f) in Fig. 3); the fluctuation excursions did not exceed 15 nT (see panel (f) in Fig. 3).

## 4.2 Level of geomagnetic field fluctuations

Up to 11:00 UT on 29 August 2019, the variations in the geomagnetic field $H$ and $D$ components in the 1–1000 s period range at the V. N. Karazin Kharkiv National University Geomagnetic Observatory, Ukraine (49.65°N, 36.93°E) were insignificant, less than 0.2–0.3 nT (Fig. 4); from 11:00 UT to 17:00 UT, their level occasionally showed increases of up to ±1 nT. On 30 August 2019, approximately in the course of the sudden storm commencement, the level of fluctuations exhibited an increase by a factor of 2 to 3 times, which persisted for about 4–5 h. On 31 August 2019, in the course of the

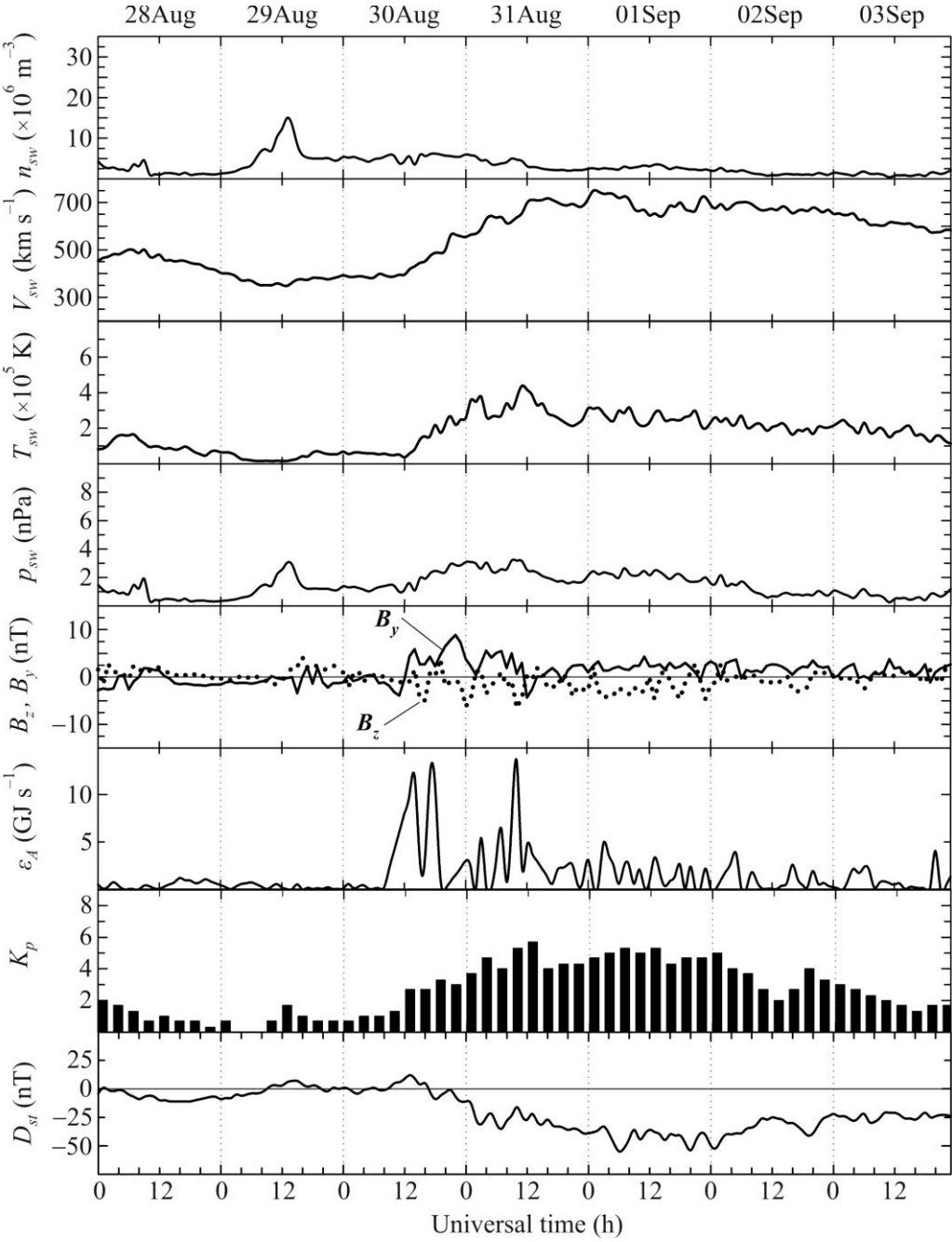

Figure 2: Universal time dependencies of the solar wind parameters: proton number density $n_{sw}$, temperature $T_{sw}$, plasma flow speed $V_{sw}$ (retrieved from https://omniweb.gsfc.nasa.gov/form/dx1.html), calculated dynamic pressure $p_{sw}$, components $B_z$ and $B_y$ of the interplanetary magnetic fields (retrieved from https://omniweb.gsfc.nasa.gov/form/dx1.html), calculated energy input per unit time, $\varepsilon_A$, from the solar wind into the Earth's magnetosphere; $K_p$- and $D_{st}$-index (retrieved from https://omniweb.gsfc.nasa.gov/form/dx1.html) for 28 August–3 September 2019 period. Dates are shown along the upper abscissa axis.

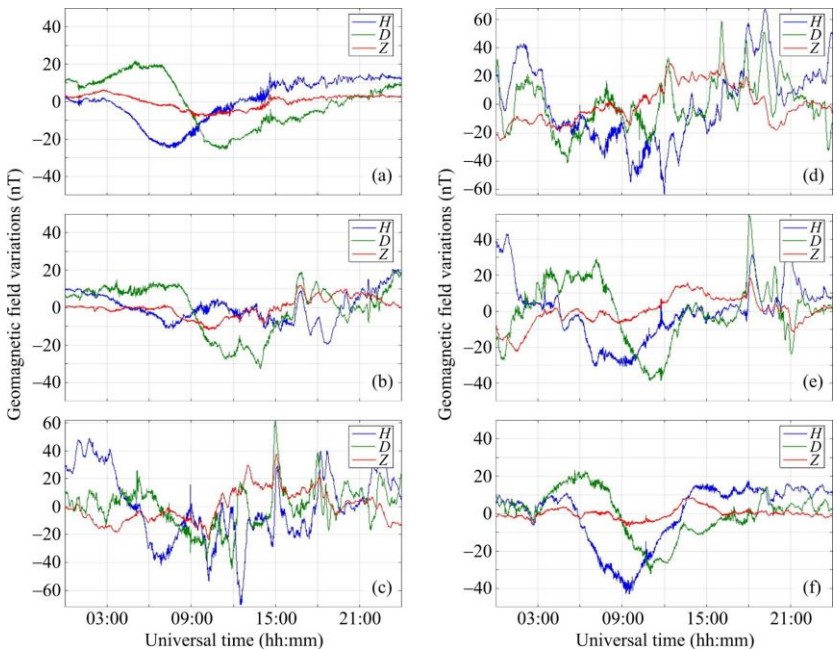

Figure 3: *H*, *D*, *Z* components for (*a*) 29 August 2019; (*b*) 30 August 2019; (*c*) 31 August 2019; (*d*) September 01, 2019; (*e*) September 02, 2019; (*f*) September 03, 2019 (retrieved from

179

180

storm main phase, the level of fluctuations showed an increase of up to 1.5–2 nT, and occasionally even of up to 4 nT. The duration of this effect was no less than 10 h.

On 1 September 2019, approximately from 08:00 UT to 13:00 UT, a considerable, of up to 2–4 nT, increase in the level of fluctuations was also observed to occur. On 2 and 3 September 2019, the level of fluctuations also exhibited occasional enhancements, of up to 1.5–2 nT, approximately 1 h in duration.

## 5 Analysis of ionospheric state

The state of the ionosphere has been analyzed in general using the data from two ionosondes. The first of these is located in the vicinity of the propagation paths used for obliquely sounding the ionosphere, viz, near the City Wakkanai (45.16°N, 141.25°E), Japan. To assess the characteristic extent of the ionospheric storm, ionosonde data from the City of Moscow (55.47°N, 37.30°E), the Russian Federation, have been used.

### 5.1 Data from ionosonde in Japan

Since 29 August 2019 to 3 September 2019, the minimum frequency, $f_{\min}$, showed insignificant variations, from 1.4 MHz to 1.5 MHz. Only on 1 September 2019, the $f_{\min}$ was observed to exhibit spikes of up to 1.7–2 MHz.

The behavior of the *E*-layer critical frequency, $f_{oE}(t)$, was observed to be approximately the same on all the days. During the daytime, this frequency attained 2.9–3.2 MHz; in the local evening, it decreased to 1.8 MHz; during night, the $f_{oE}$ was not observed, and in the course of three hours in the morning, it showed an increase from 1.8 MHz to ~3 MHz.

The sporadic-*E* critical frequency, $f_oE_s$, exhibited variations in a broad range of frequencies, from ~3 MHz to ~12–16 MHz. In the course of the storm's main phase, the $f_oE_s$ variations were insignificant.



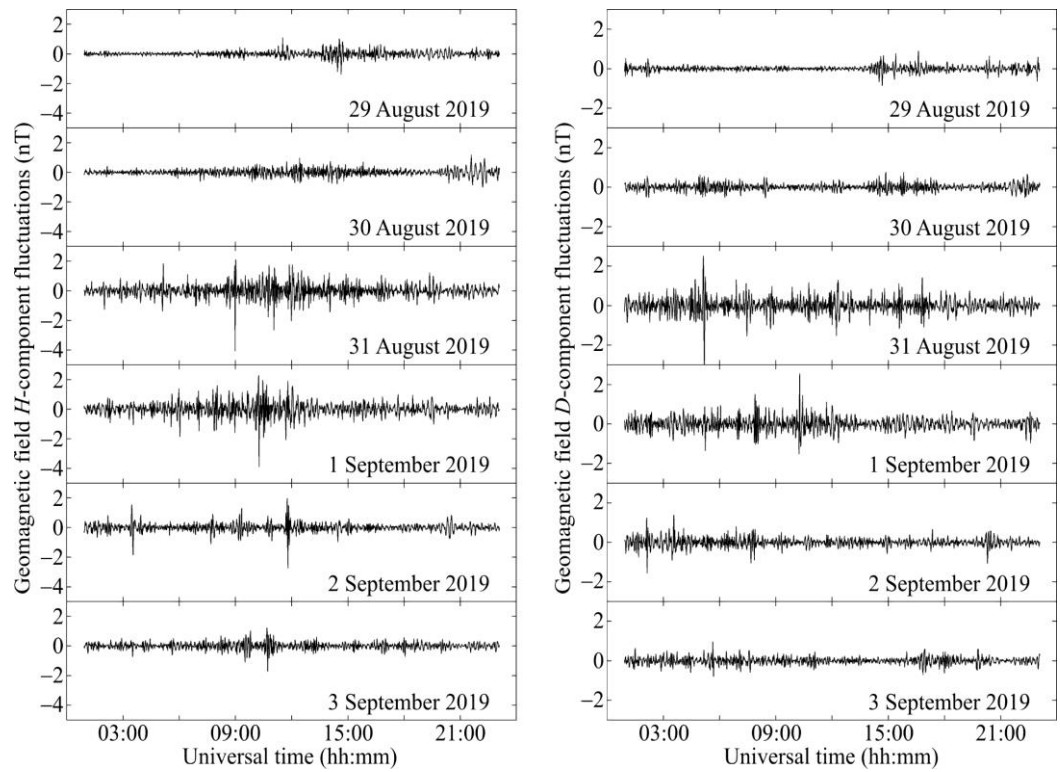

Figure 4: Magnetic field variations at V. N. Karazin Kharkiv National University Magnetometer Observatory.

Variations in the critical frequency, $f_oF_2(t)$, of the $F_2$ layer for the ordinary wave were observed to be small. During
the daytime, this frequency was observed to be approximately 5 MHz, and during night, it showed a gradual decrease from
4 MHz to 3 MHz.
Generally, the universal time variations in the virtual height, $h'_E(t)$, of the $E$ layer were observed to be
insignificant, a mere 5–10 km. However, approximately from 16:00 UT to 19:00 UT on 31 August 2019 and on 1 September
2019, the height $h'_E(t)$ showed an increase from ~100 km to ~120 km.
The sporadic $E_s$ layer virtual height exhibited considerable fluctuations, from ~80 km to 160–170 km.
We have not succeeded in obtaining reliable data on the virtual height, $h'_{F2}(t)$, of the $F_2$ layer. Most likely, it varied
from 200 km to 300 km.

**5.2 Data from ionosonde at Moscow**

The minimum frequency, $f_{min}$, values most frequently occurred in the 1.2–1.7 MHz range, and spikes of up to 2–3 MHz were
observed only sometimes. From 07:30 UT to 08:30 UT on 31 August 2019, the $f_{min}$ showed an increase from 1.4 MHz to
2.2–2.4 MHz. During 1 through 3 September 2019, the $f_{min}$ values exhibited considerable fluctuations.
The $E$-layer critical frequency, $f_{oE}(t)$, tracked the local time dependence of the electron density. The root-mean-
square $f_{oE}$ deviation did not exceed ~0.1 MHz. In the daytime, the $f_{oE}$ attained approximately 3 MHz, in the morning and in
the evening, it showed an increase or a decrease of 1.3–1.4 MHz. Under nighttime conditions, we have not succeeded in
measuring $f_{oE}$.

The sporadic-$E$ critical frequency, $f_oE_s$, exhibited considerable fluctuations, from 2 MHz to 5–7 MHz. The fluctuation excursions in $f_oE_s$ under daytime conditions were observed to be greater than under nighttime conditions.

On 31 August 2019, from 05:00 UT to 08:00 UT, the $f_{oEs}$ exhibited an increase from 3 MHz to 6–7 MHz.

The critical frequency, $f_oF_2(t)$, of the $F_2$ layer for the ordinary wave showed a decrease to 3 MHz during the 28/29 August 2019 night, which was followed by an increase to 4.5 MHz during the daytime, and even by an increase up to 5 MHz on 30 August 2019. During almost all local daytime on 31 August 2019, the $f_oF_2(t)$ was observed to be 0.7–1.1 MHz lower than on 29 August 2019. On 31 August 2019, from 09:00 UT to 11:00 UT and from 12:00 UT to 15:00 UT, an increase in $f_oF_2(t)$ was observed to be 0.7–0.8 MHz. During night and in the morning on 1 September 2019, the $f_oF_2$ values were observed to be 0.5–0.6 MHz lower than those observed on 2 September 2019; during the daytime, the difference between these frequencies did not exceeded 0.2–0.3 MHz on average.

The virtual height, $h'_E$, of the $E$ layer exhibited fluctuations in the 95–100 km range. On 31 August 2019, from 10:00 UT to 13:00 UT, it showed an increase from 102 km to 113 km. A considerable increase in $h'_E$ from 110 km to 133 km also occurred at ~12:30 UT on September 1. 2019.

The sporadic $E_s$ layer virtual height, $h'_{Es}$, exhibited fluctuations in the 100–105 km to 130–140 km range. On 31 August 2019, from 10:00 UT to 13:00 UT, this height showed an increase from ~105 km to 130 km. An increase from ~110 km to 125–132 km also took place on 1 September 2019, from 08:00 UT to 14:00  UT.

The virtual height, $h'_{F_2}$, of the $F_2$ layer exhibited significant, from ~200 km to 400–500 km, fluctuations during the 29 August to 3 September 2019 period. Sharp, from 250 km to 400–450 km, spikes in $h'_{F_2}$ took place on 31 August 2019, during 13:30–14:30 UT and 16:00–16:30 UT periods. Considerable, from 250–300 km to 400–500 km, variations in $h'_{F_2}$ were also observed to occur during the 31 August 2019 to 1 September 2019 night, as well as from 16:00 to 18:00 UT on 1 September 2019.

## 6 Ionosphere: Oblique incidence sounding

### 6.1 Lintong/Pucheng to Harbin radiowave propagation path

The radio station operating at 5,000 kHz is located in the People's Republic of China at a great-circle propagation path range, $R$, of 1,875 km from the receiver.

Approximately from 00:00 UT to 07:00 UT on 29 August 2019, i.e., during sunlit hours on the reference day, the signal amplitude, $A$, was observed to be ~–70 dBV, and the Doppler shift of frequency in the main ray signal, $f_D(t)$, to be ~0. 0 Hz, as can be seen in Fig. 5. After sunset at ~07:00 UT, i.e., in the evening hours, the $A$ showed a gradual increase of up to –40 dBV. The $f_D(t)$ values gradually decreased from 0 Hz to –(0.5–1) Hz. Approximately from 09:00 UT to 16:00 UT, the Doppler spectra were observed to significantly broaden, from –2.5 Hz to 2 Hz. On 30 August 2019, the $f_D(t)$ exhibited considerable, from –0.3 Hz to 0.4 Hz, variations during the 18:00 UT to 22:00 UT period.

On 31 August 2019, the $f_D(t)$ changed from –0.3 Hz to 0.3 Hz over the 12:00–18:00 UT period when quasi-periodic variations in the $f_D(t)$ took place with ~40 min period, $T$, and ~0.20–0.25 Hz amplitude, $f_{Da}$. From 17:00 UT to 22:00 UT, the amplitude $A(t)$ exhibited considerable, up to 15–20 dBV, variations.

On 1 September 2019, the $f_D(t)$ showed significant increase, from –1.8 Hz to 1.4 Hz, in the course of sunset in the ionosphere. The ionospheric storm effect was observed to occur from at least 10:00 UT to 19:00 UT. The amplitude $A(t)$ was observed to exhibit considerable, up to 20 dBV, variations during the 11:30–21:00 UT period. On 2 and 3 September 2019, the behavior of the Doppler spectra almost did not differ from that on the undisturbed day.

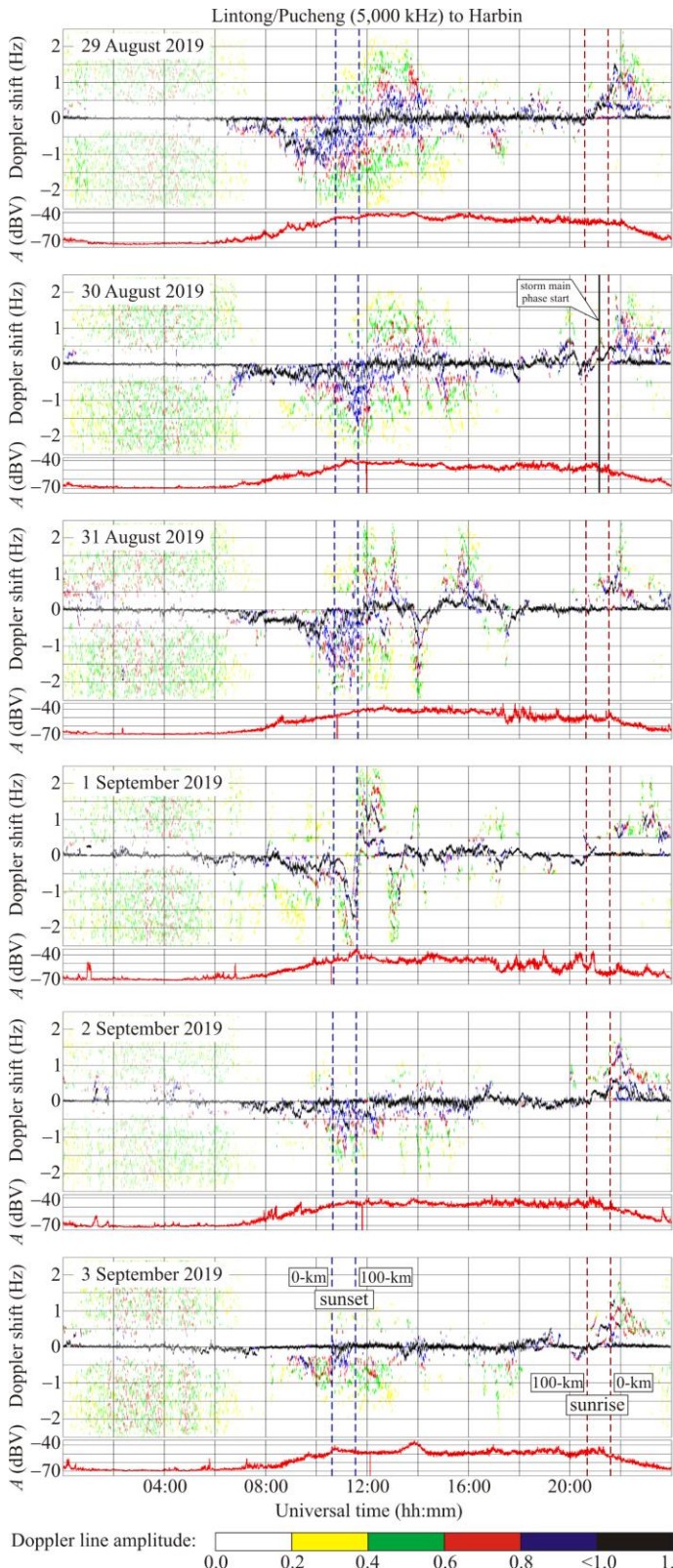

Figure 5: Universal time variations of Doppler spectra and relative signal amplitude, *A*, along the Lintong/Pucheng to Harbin propagation path for 29–31 August 2019 and 1–3 September 2019 (panels from top to bottom). The Doppler shift plot is comprised of 117,600 samples in every 1 h interval. The signal amplitude, *A*, at the receiver output in decibels, dBV, relative to 1 V is shown below the Doppler spectrum in every panel. The dashed lines indicate the sunsets and sunrises at 0- and 100-km altitude.

## 6.2 Hwaseong to Harbin radiowave propagation path

The 6,015 kHz transmitter is located in the Republic of Korea at an ~950 km distance from the receiver, and it did not operate from 00:00 UT to 03:40 UT.

On 29 August 2019, the Doppler shift of frequency $f_D(t) \approx 0$ Hz at almost all times (Fig. 6). The spectra were observed to exhibit maximum broadening near the dawn and dusk terminators. The variations in the signal amplitude represented the local time behavior.

On 30 August 2019, considerable (from –0.4 Hz to 0.4 Hz) variations in the Doppler shift of frequency in the main ray were observed to occur from 13:00 UT to 21:00 UT with an ~70–110 min quasi-period, $T$, and an ~0.4 Hz amplitude, $f_{Da}$.

On 31 August 2019, quasi-periodic changes in $f_D(t)$ were observed to occur from 12:00 UT to 17:00 UT with $T \approx 40$ min and $f_{Da} \approx 0.4$–0.7 Hz.

On 1 September 2019, very significant (from –1.5 Hz to 1.3 Hz) variations in $f_D(t)$ and the Doppler spectra took place from 10:00 UT to 14:00 UT and from 16:30 UT to 19:00 UT. From approximately 10:00 UT to 21:00 UT, large (up to 30 dBV) variations in signal amplitudes were evident.

On 2 and 3 September 2019, the Doppler spectra and signal amplitudes did not exhibit considerable variations.

## 6.3 Chiba/Nagara to Harbin radiowave propagation path

The radio station operating at 6,055 kHz is located in Japan at an ~1,610 km range from the receiver. The signal transmissions were absent from 15:00 UT to 22:00 UT.

The Doppler spectra exhibited similar behavior on 29, 30, and 31 August 2019 (Fig. 7). From 06:00 UT to 15:00 UT, the spectra were observed to be spread; they occupied the –1.5 Hz to 1.5 Hz frequency range.

On 1 September 2019, the Doppler spectra exhibited behavior sharply different from that observed on the preceding day. The spread was evident weakly; from 10:00 UT to 15:00 UT, the Doppler shifts of frequency exhibited sharp changes from –1.5 Hz to 1.3 Hz; the quasi-periodic process with the ~60 min and greater period, $T$, and the ~0.2 Hz and greater amplitude, $f_{Da}$, became evident. On this day, the signal amplitude also exhibited considerable (up to 20 dBV) fluctuations.

On 2 September 2019, the Doppler spectra remained still disturbed over the 07:00–12:00 UT period.

On 3 September 2019, the Doppler spectrum spread was insignificant. The Doppler shift of frequency, $f_D(t)$, was observed to be close to zero level most of the time.

## 6.4 Beijing to Harbin radiowave propagation path

The 6,175 kHz transmitter is located in the People's Republic of China at approximately 1,050 km range from the receiver. The transmitter operated only over the 09:00 UT to 18:00 UT and 20:20 UT to 24:00 UT periods.

On 29 and 30 August 2019, the Doppler spectra were characteristic of the single ray propagation; the second ray appeared only sporadically (Fig. 8). The Doppler shift of frequency, $f_D(t)$, was observed to be close to zero level almost all the time, and the signal amplitude $A(t) \approx -15$ dBV.

On 31 August 2019, over the 12:00–18:00 UT period, the behavior of $f_D(t)$ sharply changed. The $f_D(t)$ dependence became quasi-periodic with an ~30 min period, $T$, and an ~0.2 Hz amplitude. At approximately 14:00 UT, the $f_D$ dependence exhibited a sharp decrease from 0.2 Hz to –0.7 Hz.

The $f_D$ was observed to exhibit considerable, from –1.2 Hz to 1.1 Hz, variations over the 10:00–12:00 UT and 16:00–18:00 UT periods on 1 September 2019, while the signal amplitude showed a decrease by 30 dBV from 16:00 UT to 18:00 UT.

On 2 and 3 September, 2019, the Doppler spectra exhibited the behavior characteristic of the quiet ionosphere.

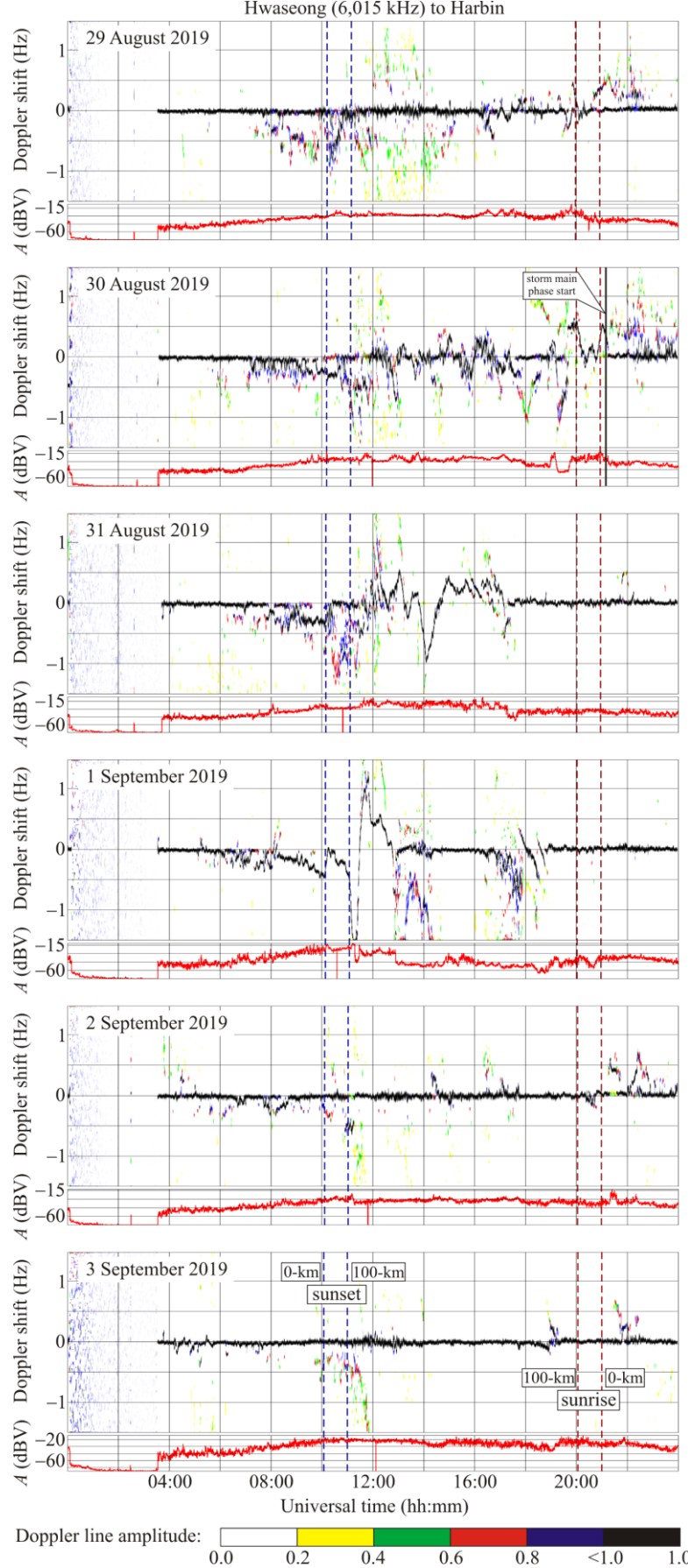

Figure 6: Same as Figure 5, but for the Hwaseong to Harbin radiowave propagation path at 6,015 kHz.

300

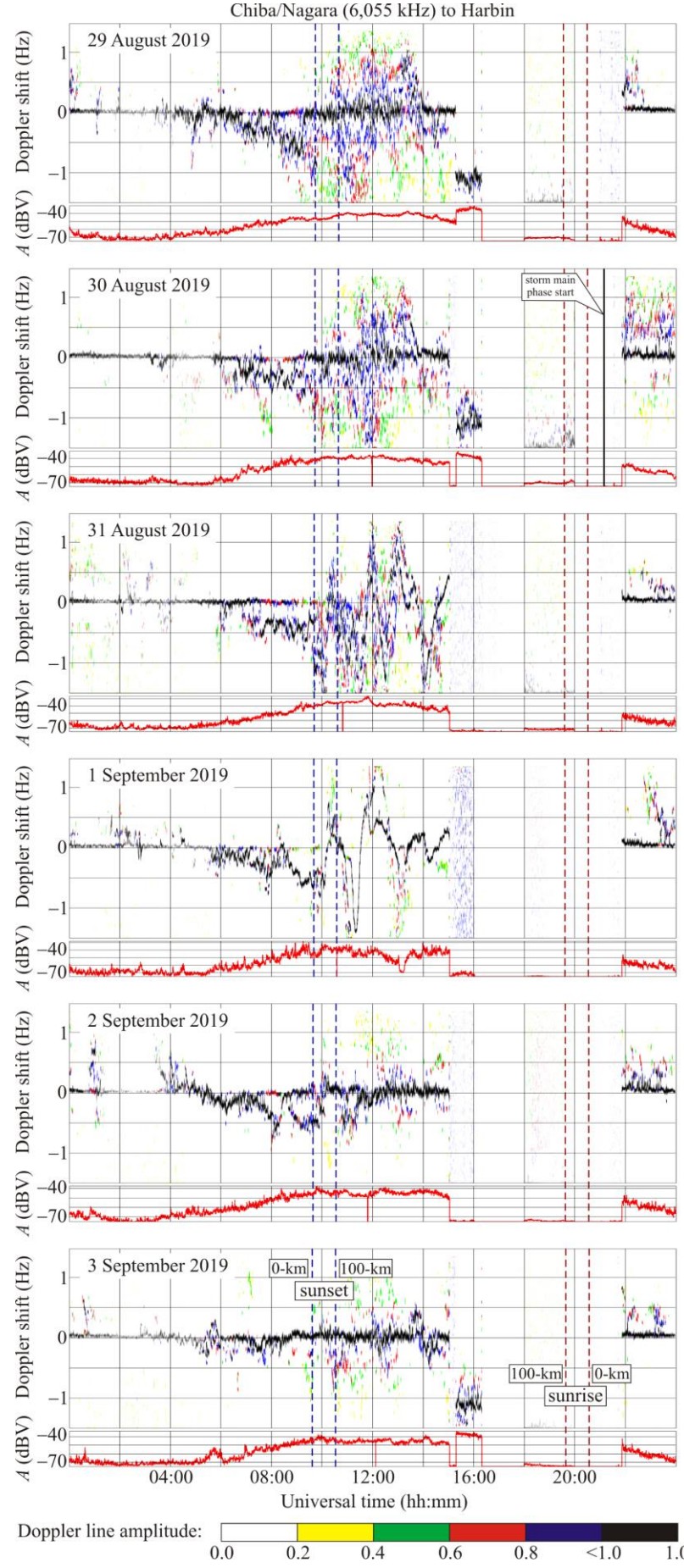

Figure 7: Same as Figure 5, but for the Chiba/Nagara to Harbin radiowave propagation path at 6,055 kHz.

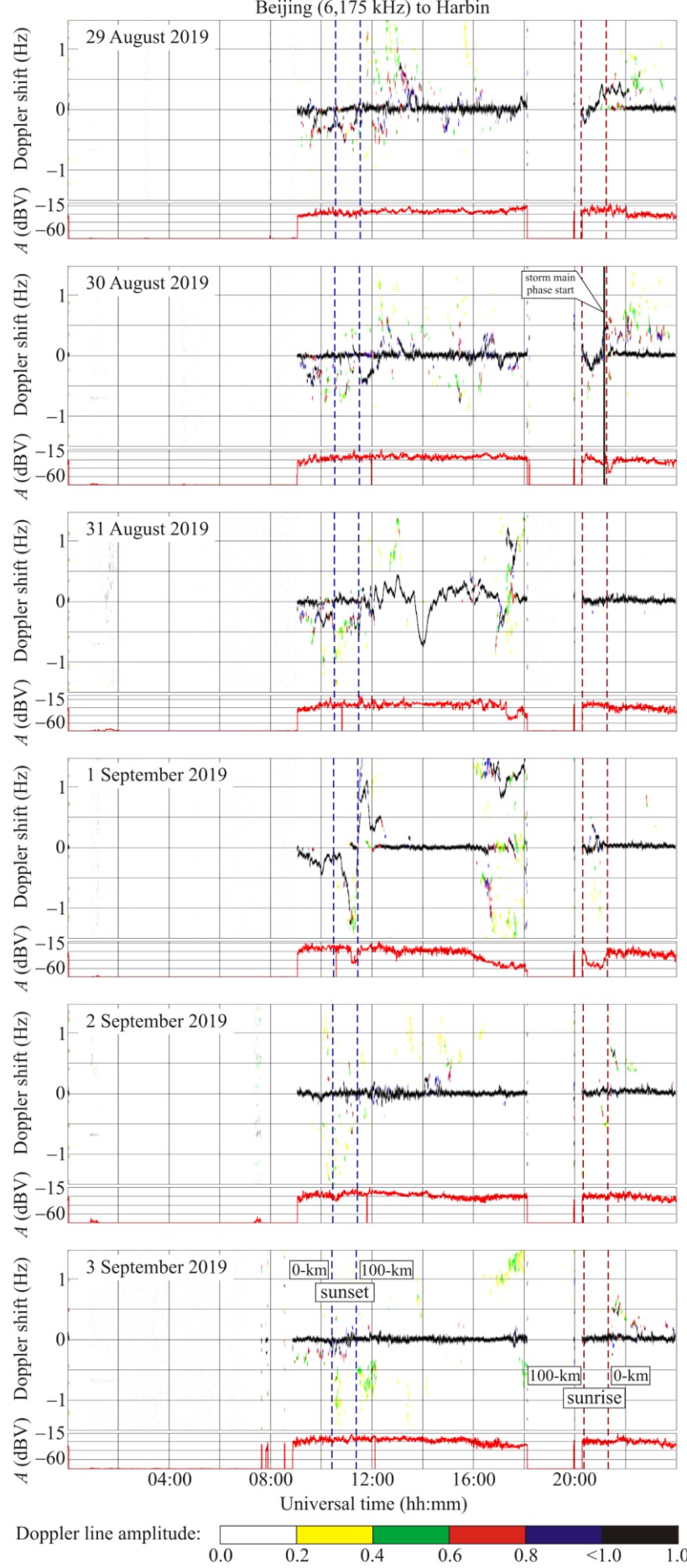

Figure 8: Same as Figure 5, but for the Beijing to Harbin radiowave propagation path at 6,175 kHz.

## 6.5 Goyang to Harbin radiowave propagation path

The radio station operating at 6,600 kHz is located in the Republic of Korea at a range, $R$, of ~910 km from the receiver. From 05:00 UT to 08:50 UT, the Doppler measurements were not possible over the entire measurement interval, and on 3 September 2019, during 10:00 – 11:30 UT period.

On 29 August 2019, the Doppler spectra represented the undisturbed state of the ionosphere. For the main ray, the Doppler shift of frequency $f_D(t) \approx 0$ Hz (Fig. 9).

On 30 August 2019, from 09:00 UT to 14:00 UT, the Doppler spectra showed a noticeable broadening. Over the same time period, the signal amplitude experienced an enhancement in fluctuations, attaining 15–20 dBV.

On 31 August 2019, from 09:00 UT to 17:00 UT, considerable, from –1.3 Hz to 0.7 Hz, variations took place in the Doppler shift of frequency, $f_D(t)$. The variations in $f_D(t)$ were observed to be quasi-periodic, with ~40 min periods, $T$, and ~0.2–0.5 Hz amplitudes, $f_{Da}$. From 17:30 UT to 19:00 UT, $T \approx 15$ min, and $f_{Da} \approx 0.1$ Hz; the signal amplitude exhibited sporadic changes of up to 30 dBV.

On 1 September 2019, over the 08:30–13:00 UT period, the $f_D(t)$ also showed significant variations, from –1.5 Hz to 0.7 Hz. The signal amplitude, $A(t)$, fluctuated wildly, up to 30 dBV.

On 2 and 3 September 2019, the $f_D(t)$ and $A(t)$ showed virtually no change. The state of the ionosphere along the propagation path was quiet.

## 6.6 Ulaanbaatar to Harbin radiowave propagation path

The radio station operating at 7,260 kHz is located in Mongolia at an ~1,496 km range from the receiver. It was switched off from 05:00 UT to 07:00 UT and from 18:00 UT to 20:30 UT.

On 29 August 2019, the Doppler spectra showed that the propagation was more likely to occur along a single ray, the $f_D(t)$ varied virtually monotonically (Fig. 10).

On 30 August 2019, from 12:00 UT to 15:00 UT, the $f_D(t)$ exhibited quasi-periodic variations with 20 and 40 min periods, $T$, and with an ~0.1 Hz amplitude, $f_{Da}$, for $T \approx 20$ min and with $f_{Da} \approx 0.3$ Hz for $T \approx 40$ min.

On 31 August 2019, the $f_D(t)$ fluctuated wildly and varied quasi-periodically with an ~20 min period, $T$, and an ~0.1 Hz amplitude, $f_{Da}$, almost all the time; from 13:30 UT to 14:00 UT, it exhibited a sharp decrease from 0 Hz to –1.5 Hz, which was followed by a subsequent increase from –1.5 Hz to 0 Hz.

On 1 September 2019, during the 09:00–12:30 UT period, sharp changes in $f_D(t)$ became evident, from 0 Hz to –1.5 Hz and conversely.

On 2 September 2019, from 11:00 UT to 15:00 UT, the $f_D(t)$ exhibited quasi-peiodic variations with an ~20–25 min period, $T$, and an ~0.1 Hz amplitude, $f_{Da}$.

On 3 September 2019, from 13:00 UT to 15:00 UT, quasi-peiodic variations in $f_D(t)$ with an ~60 min period, $T$, and an ~0.15 Hz amplitude, $f_{Da}$, were also observed to occur.

Since 30 August 2019 through 2 September 2019, an increase in the frequency and level of fluctuations in signal amplitude were noted.

## 6.7 Yakutsk to Harbin radiowave propagation path

The 7,350 kHz transmitter is located in the Russian Federation at a range, $R$, of ~1,845 km from the receiver. Unfortunately, the transmitter operated only over the 11:00–18:00 UT and 20:15–24:00 UT periods.

On 29 and 30 August 2019, the Doppler spectra and signal amplitude exhibit relatively small variations (Fig. 11).

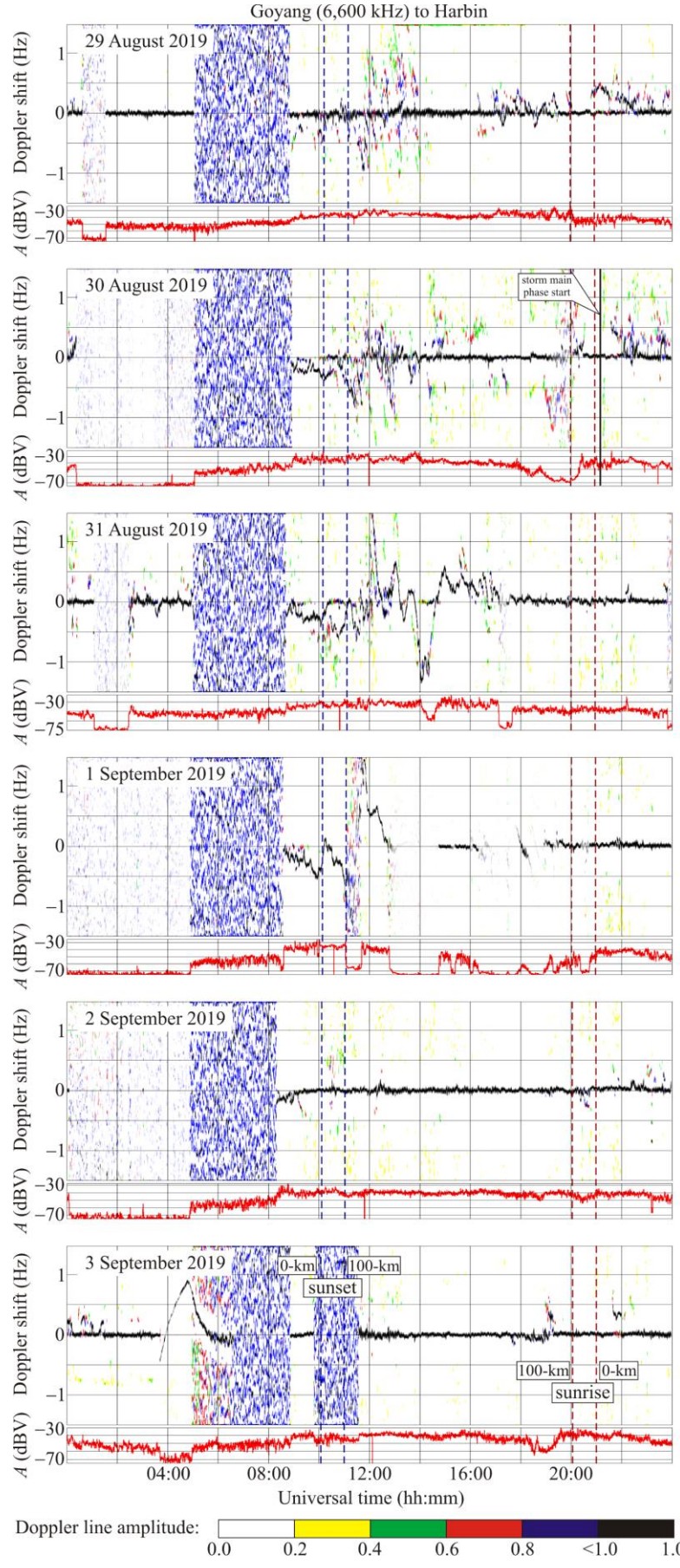

Figure 9: Same as Figure 5, but for the Goyang to Harbin radiowave propagation path at 6,600 kHz.


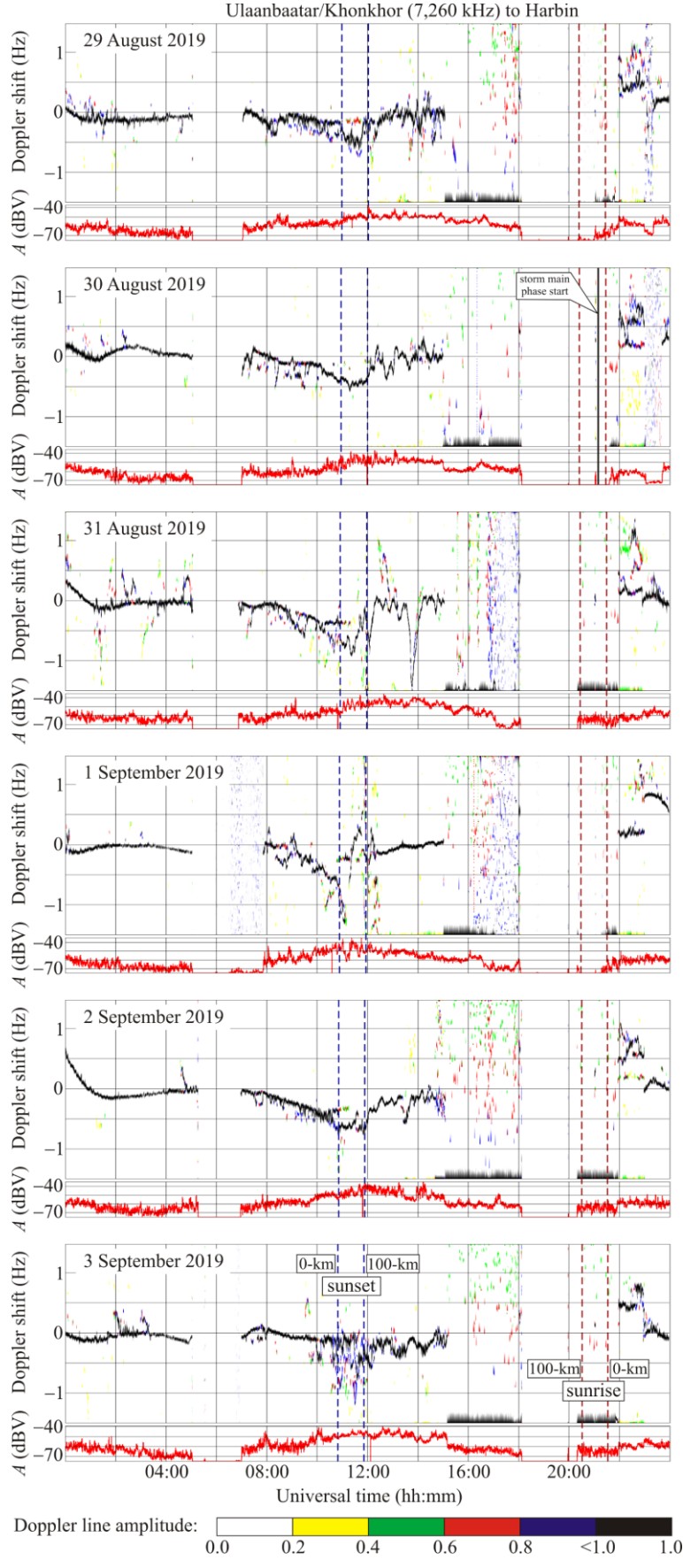

Figure 10: Same as Figure 5, but for the Ulaanbaatar/Khonkhor to Harbin radiowave propagation path at 7,260 kHz.


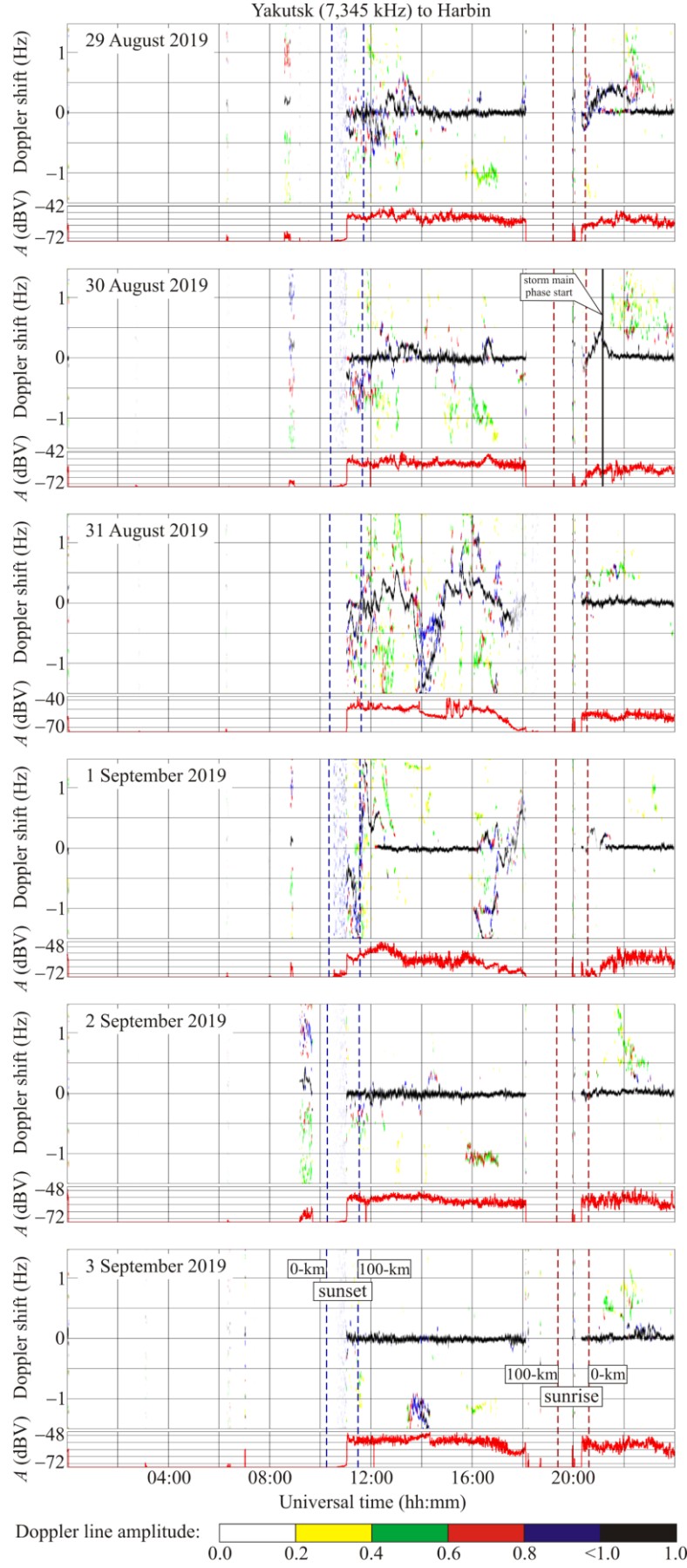

Figure 11: Same as Figure 5, but for the Yakutsk to Harbin radiowave propagation path at 7,345 kHz.


On 31 August 2019, the Doppler spectra occupied the –1.5 Hz to 1.5 Hz range. The $f_D(t)$ varied quasi-periodically

with an ~24 min period, $T$, and ~0.2 Hz amplitude, $f_{Da}$. From 13:40 UT to 14:50 UT, the $f_D(t)$ exhibited a decrease in $f_D(t)$
from 0 Hz to –1.5 Hz, which was followed by an increase from –1.5 Hz to 0 Hz, while the amplitude showed a decrease by
10 dBV. From 15:00 UT to 16:00 UT, the excursion of fluctuations in $A(t)$ attained 20 dBV.

On 1 September 2019, the Doppler spectra and the signal amplitudes exhibited considerable variations during the

11:00–13:00 UT and 16:00–18:00 UT periods. From 16:00 UT to 18:00 UT, the spectra varied quasi-periodically with 30–40
min periods, $T$, and 0.15 Hz amplitudes, $f_{Da}$.

On 2 and 3 September 2019, the behavior of $f_D(t)$ and $A(t)$ represented the behavior of the quiet ionosphere.

**6.8 Shijiazhuang to Harbin radiowave propagation path**
The radio station operating at 9,500 kHz is located in the People's Republic of China at an ~1,310 km range, $R$, from the
receiver.

On 29 and 30 August 2019, the behaviors of the Doppler spectra and signal amplitudes were similar. The

ionosphere did not experience appreciable disturbances (Fig. 12).

On 31 August 2019, the Doppler spectra showed that the propagation is more likely to occur along a single ray. The

$f_D(t)$ exhibited significant variations, from –1 Hz to 0.8 Hz. Quasi-periodic variations in $f_D(t)$ with an ~30 min period, $T$, and
an ~0.3–0.5 Hz amplitude, $f_{Da}$, became evident. From 17:00 UT to 20:25 UT, $A(t) \approx$ –70 dBV, the signal amplitude was
observed to be at the noise level. On 1 September 2019, the signal amplitude was also observed to be at the noise level
during the 09:10–11:50 UT and 17:00–21:40 UT periods; during the rest of the time, $f_D(t) \approx 0$ Hz.

The behavior of the Doppler spectra and the signal amplitudes on 2 and 3 September, 2019 was characteristic of the

undisturbed state of the ionosphere. Since $f_D(t) \approx 0$ Hz all the time, the radio wave was apparently reflected from the $E_s$ layer
screening the ionospheric $F$ region.
**6.9 Hohhot to Harbin radiowave propagation path**
The 9,520 kHz transmitter is located in the People's Republic of China at an ~1,340 km range from the receiver. The radio
station usually does not broadcast from 16:00 UT to 21:40 UT.

On 29 August 2019, considerable variations in the Doppler spectra, $f_D(t)$, and the signal amplitude, $A(t)$, were

observed to occur near the dusk and dawn terminators in the ionosphere (Fig. 13).

On 30 August 2019, significant variations in the Doppler spectra became evident from 14:00 UT to 16:00 UT.

On 31 August 2019, considerable, from –0.7 Hz to 0.7 Hz, variations in $f_D(t)$ took place over the 11:00–13:30 UT

period. The period, $T$, is observed to be ~24 min, and the amplitude, $f_{Da}$, ~0.1–0.5 Hz.

On 1 September 2019, $f_D(t) \approx 0$ Hz almost all the time. Significant, 20–40 dBV, variations in $A(t)$ were observed to

occur from 08:00 UT to 16:00 UT.

On 2 and 3 September 2019, the ionosphere did not experience considerable disturbances.

**6.10 Yamata to Harbin radiowave propagation path**
The 9,750 kHz transmitter is located in Japan at an ~1,570 km range, $R$, from the receiver. The transmissions are usually
absent from 16:00 UT to 22:00 UT.

During the local daytime on 29–31 August 2019, the Doppler shift of frequency usually fluctuated around ~0 Hz

with periods, $T$, of about 20–30 min and amplitudes, $f_{Da}$, of about 0.1 Hz (Fig. 14). From 10:00 UT to 14:00 UT, the Doppler
spectra exhibited a significant broadening, and the $f_D(t)$ showed chaotic behavior.

On 30 August 2019, from 12:00 UT to 16:00 UT, the signal amplitude, $A(t)$, exhibited near-quasi-periodic

variations with a period, $T$, of about 30 min and 10–15 dBV excursions.

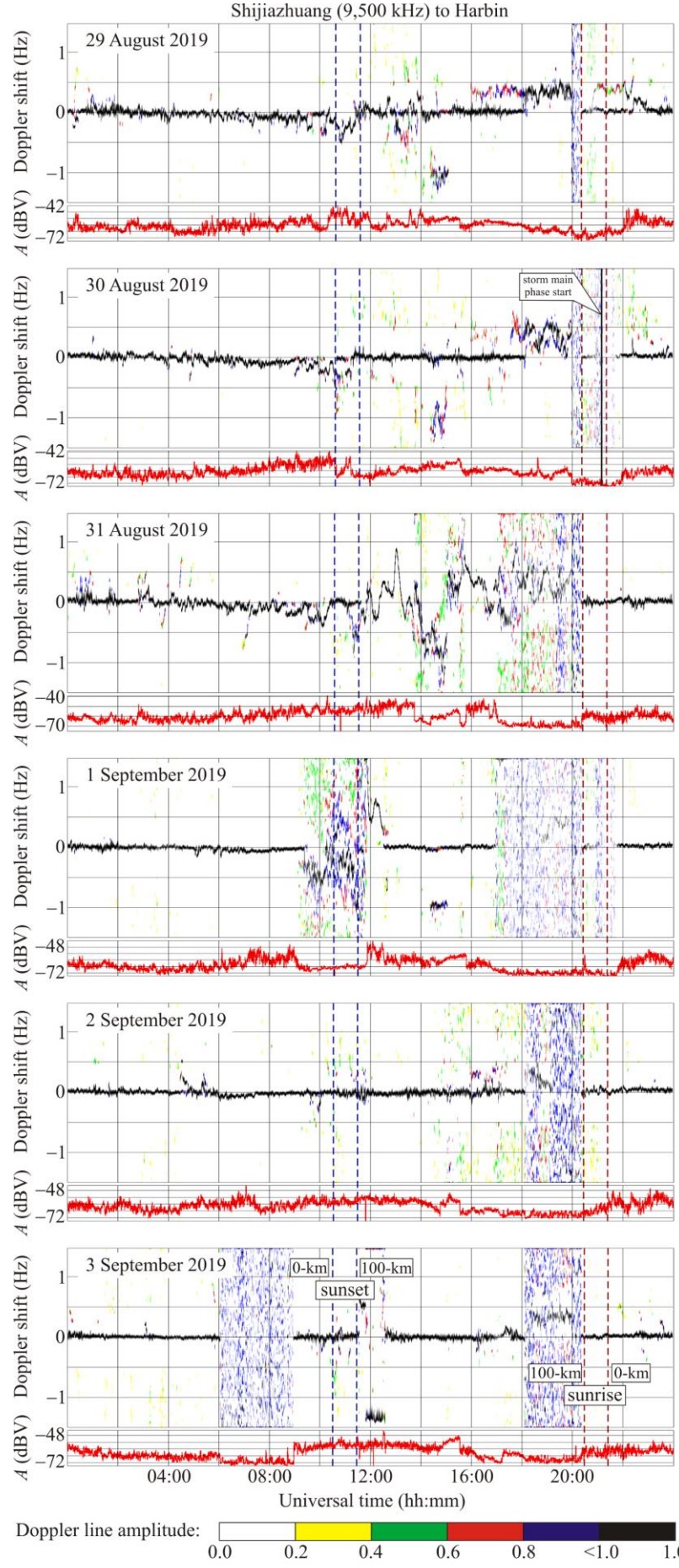

Figure 12: Same as Figure 5, but for the Shijiazhuang to Harbin radiowave propagation path at 9,500 kHz.

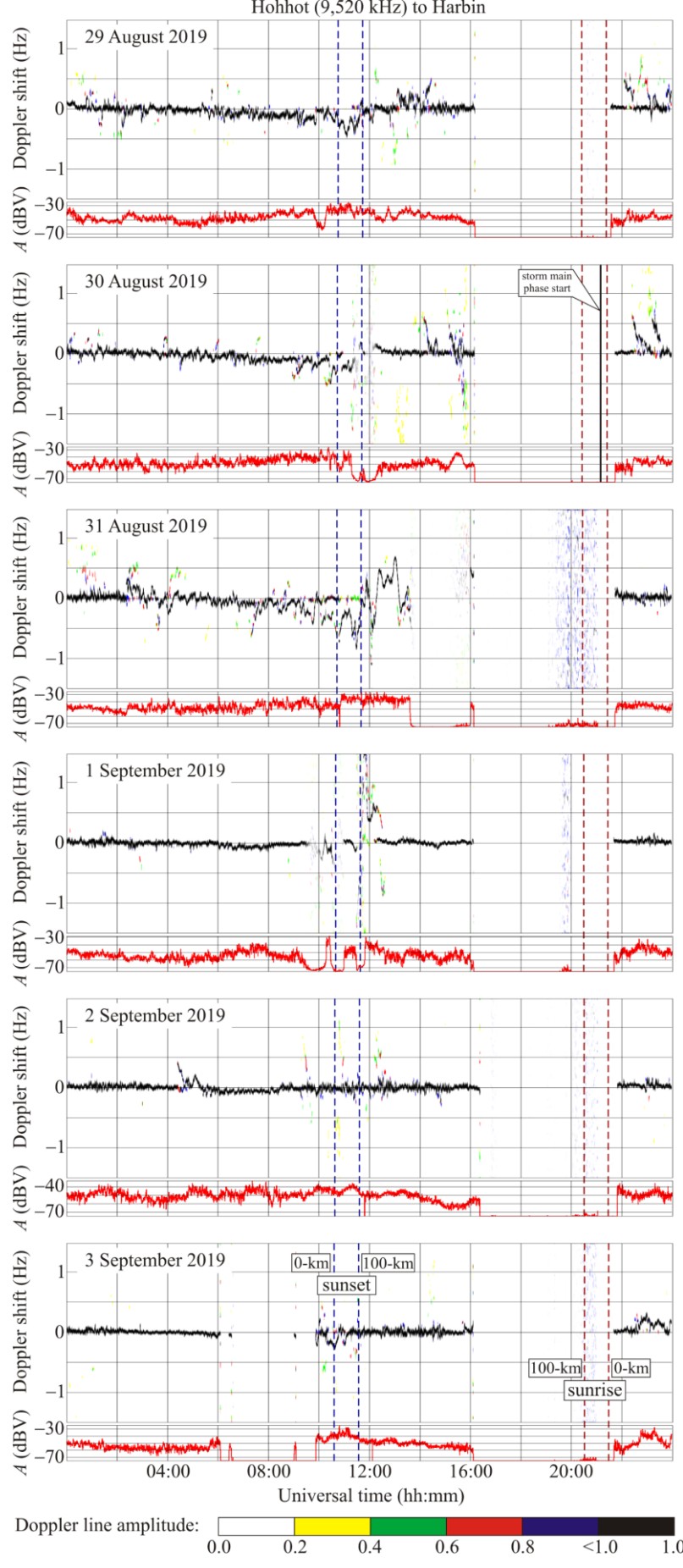

Figure 13: Same as Figure 5, but for the Hohhot to Harbin radiowave propagation path at 9,520 kHz.

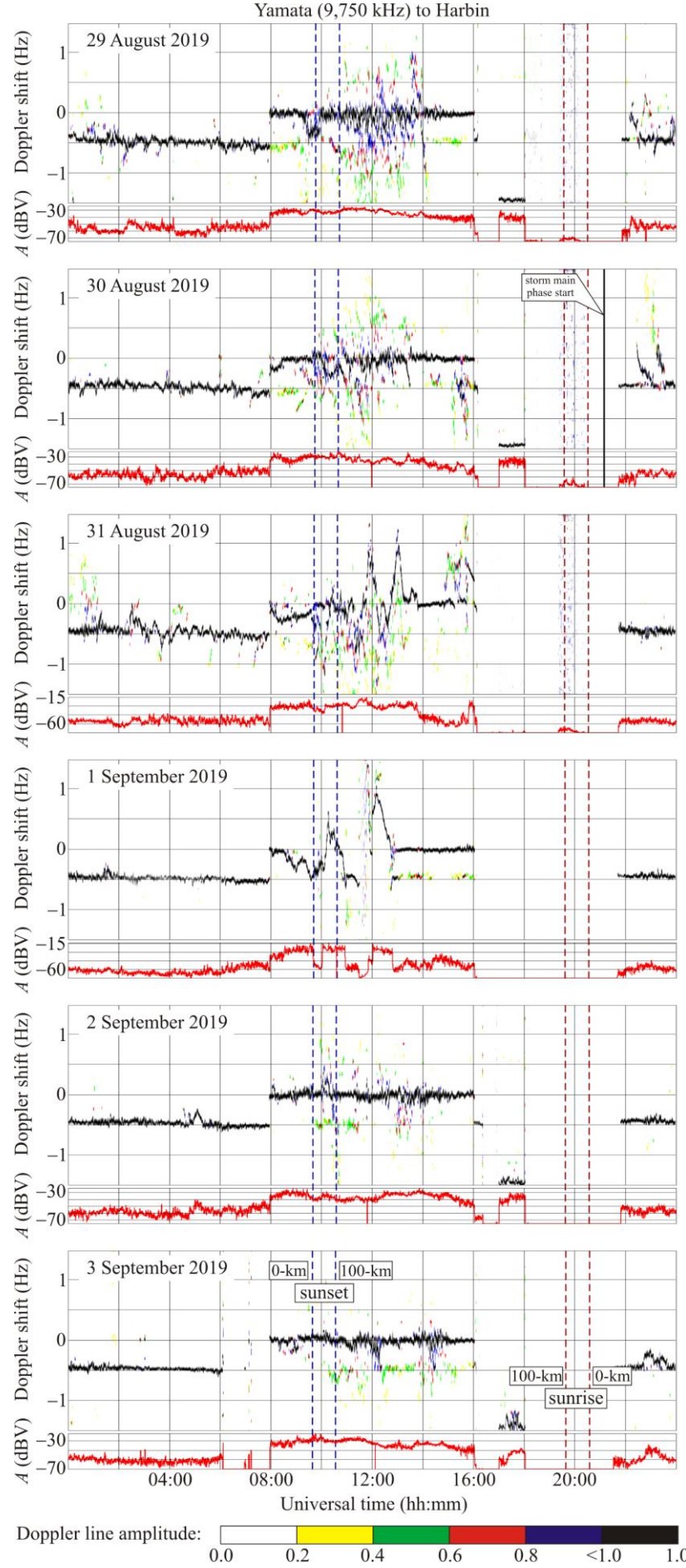

Figure 14: Same as Figure 5, but for the Yamata to Harbin radiowave propagation path at 9,750 kHz.

On 31 August 2019, a considerable, from –0.4 Hz to 0.8 Hz, increase of variations in $f_D(t)$ was observed to occur

from 12:00 UT to 16:00 UT, while the fluctuations in the signal amplitude, $A(t)$, were small, in the 10–15 dBV range.

On 1 September 2019, the excursions in $f_D(t)$ varied from –0.5 Hz to 1 Hz during the 08:00–13:00 UT period, while

the signal amplitude exhibited sharp changes, by 40–60 dBV.

On 2 and 3 September 2019, the $f_D(t)$ and $A(t)$ exhibited behavior characteristic of the quiet days.

**6.11 Beijing to Harbin radiowave propagation path**
The radio station broadcasting at 9,830 kHz over an interval shorter than half of a day is located in the People's Republic of
China at an ~1,050 km range, $R$, from the receiver.

On 29 and 30 August 2019, and on 2 and 3 September 2019, the Doppler spectra did not exhibit considerable

variations (Fig. 15). Their variations were observed to occur from 11:00 UT to 16:00 UT on 31 August 2019 and from 10:00
UT to 12:30 UT on 1 September 2019.

On 30 and 31 August 2019 and on 1 September 2019, the signal amplitude exhibited considerable, up to 30 dBV,

variations. The reflected signal was absent from 14:00 UT to 18:00 UT on 31 August 2019 and from 09:00 UT to 12:10 UT
on 1 September 2019.
**7 Discussion**
The strength of geospace storms is conveniently estimated by the energy entering the magnetosphere from the solar wind per
unit of time, the Akasofu function. The index
$$G_{st} = 10 \lg \frac{\varepsilon_A}{\varepsilon_{A\min}},$$
where $\varepsilon_{A\min} = 10$ GJ s$^{-1}$, have been introduced in (Chernogor and Domnin, 2014) and is used to measure the storm strength.
Substituting $\varepsilon_{A\max} \approx 15$ GJ s$^{-1}$ for the storm under study gives $G_{st} \approx 1.8$. According to the classification of Chernogor and
Domnin (2014), this storm is minor. Assuming the storm length to be $\Delta t \approx 10^5$ s, the energy entering the magnetosphere is
found to be $E_{st} \approx 1.5 \times 10^{15}$ J. Such a storm falls into the Geospace Storm Index 1 (GSSI1) type (Chernogor and Domnin,

2014).

**7.1 Geomagnetic field effects**
The effects in the geomagnetic field began to appear after 12:00 UT on 30 August 2019. Considerable effects in the
geomagnetic field occurred during the main phase of the magnetic storm, i. e., on 31 August 2019 and 1 September 2019.
The recovery phase persisted for 2–3 days since 00:00 UT on 2 September 2019.

Let us estimate the magnetic storm energy $E_{ms}$ and the power $P_{ms}$, using the relation of Gonzalez et al. (1994):

$$E_{ms} = \frac{3}{2} E_m \frac{|D_{st}^*|}{B_0},$$
where $B_0 \approx 3 \times 10^{-5}$ T is the equatorial magnetic induction, and $E_m \approx 8 \times 10^{17}$ J is the total energy in the Earth's dipole
magnetic field.

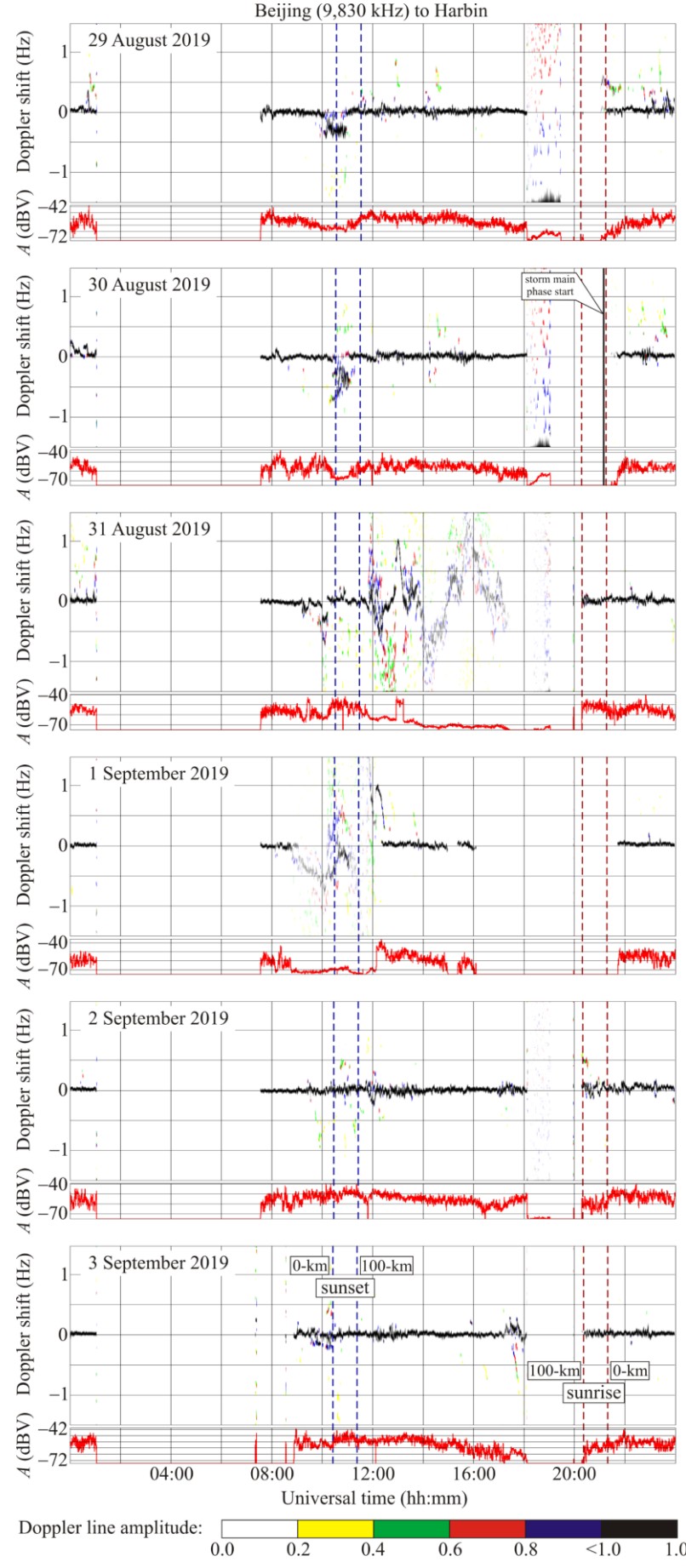

Figure 15: Same as Figure 5, but for the Beijing to Harbin radiowave propagation path at 9,830 kHz.

The corrected value of $D_{st}^{*}$ is given by
$D_{st}^{*} = D_{st} - bp_{sw}^{1/2} + c$ ,
where $b = 5 \times 10^5$ nT (J·m$^{-3}$)$^{-1/2}$, $c = 20$ nT, $p_{sw} = n_p m_p V_{sw}^2$, $m_p$ and $n_p$ are proton mass and number density, $V_{sw}$ is the solar
wind bulk speed. Given $p_{sw\max} \approx 3$ nPa, $D_{st\min} \approx -55$ nT, and $D_{st}^{*} = -62$ nT, the magnetic storm energy $E_{ms} = 1.5$ PJ. For the
magnetic storm of $1.7 \times 10^5$ s duration, the power $P_{ms} \approx 9$ GW.

In accordance with the NOAA Space Weather Scale [http://www.sec.noaa.gov], this storm is classified as moderate.

In accordance with the classification system of Chernogor and Domnin (2014), magnetic storms with $K_p = 5.0$–$5.9$ are
classified as moderate, and their energy and power lie within the $E_{ms} \approx (1$–$5) \times 10^{15}$ J and $P_{ms} \approx (6$–$22) \times 10^{10}$ W limits,
respectively.
**7.2 Effects in geomagnetic field fluctuations**
The universal time dependences of the horizontal components of the geomagnetic field in the 100–1000 s period range were
subjected to the systems spectral analysis in the 100–1000 s period range.

The results of the spectral analysis for 29 August 2019, which could be considered as reference date, are presented

in Fig. 16. The *H*- and *D*-component levels did not exceed 2–3 nT, while the spectra exhibited predominantly 600–900 s
period oscillations.

On 31 August 2019, the day when the storm's main phase was observed, the *H*- and *D*-components attained 5–

10 nT (Fig. 17). The spectra of the *H*- and *D*-components showed predominantly 300–400 s, 700–900 s and 400–600 s, 700–
900 s period oscillations, respectively.

On 1 September 2019, the levels of the components remained the same as those on 31 August 2019. The 800–1000

s period oscillations were predominant in both components.
**7.3 Ionospheric storm effects**
**7.3.1 Disturbances in ionogram parameters**
Variations in ionogram parameters observed with the Japan and Russian Federation ionosondes exhibit similar behaviors.
This suggests that the ionospheric storm under study is a large-scale phenomenon.

The list of the main effects that accompanied the ionospheric storm include the following.

1. An increase in $f_{\min}$ from 1.4 MHz to 2.2–2.4 MHz from 07:30 UT to 08:30 UT on 31 August 2019.
2. An increase in $f_{oEs}$ from 3 MHz to 6–7 MHz from 05:00 UT to 08:00 UT on 31 August 2019.
3. A decrease in $f_{oF2}$ by 0.7–1.1 MHz 31 August 2019 as compared to $f_{oF2}$ on 29 August 2019.
4. A decrease in $f_{oF2}$ by 0.2–0.6 MHz on 1 September 2019 as compared to $f_{oF2}$ on 2 September 2019.
5. An increase in $h_E'$ from 102 km to 113 km from 10:00 UT to 13:00 UT on 31 August 2019.
6. An increase in $h_E'$ from 110 km to 133 km at approximately 12:30 UT on 1 September 2019.
7. An increase in $h_{Es}'$ from 105 km to 130 km from 10:00 UT to 13:00 UT on 31 August 2019.
8. An increase in $h_{Es}'$ from 110 km to 125–132 km from 08:00 UT to 14:00 UT on 1 September 2019.
9. Brief spikes in $h_{F2}'$ from 250 km to 400–450 km from 13:30 UT to 14:30 UT and from 16:00 UT to 16:30 UT on 31
August 2019.
10. An increase $h_{F2}'$ from 250–300 km to 400–500 km during the 31 August 2019/1 September 2019 night, as well as from
16:00 UT to 18:00 UT on 1 September 2019.

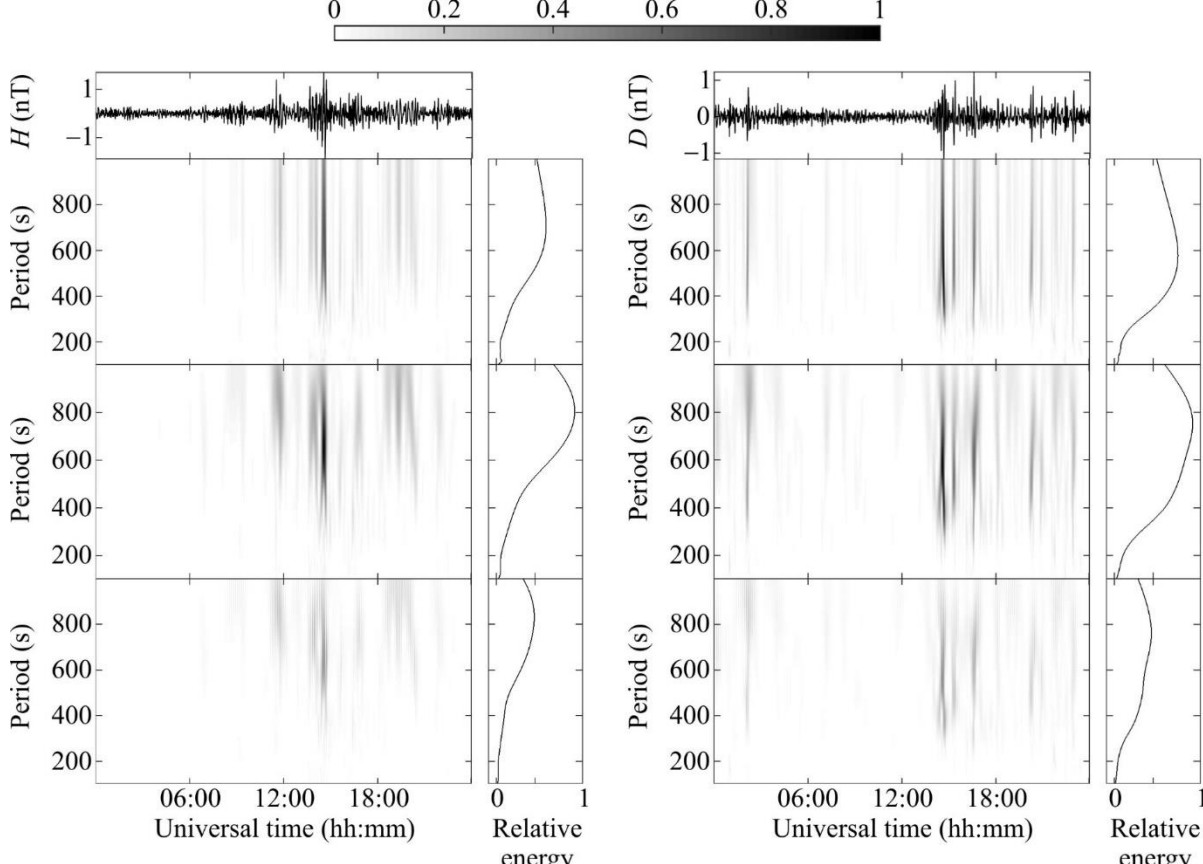


Figure 16: Systems spectral analysis products for the geomagnetic variations on 29 August 2019 at V. N. Karazin Kharkiv
National University Magnetometer Observatory.

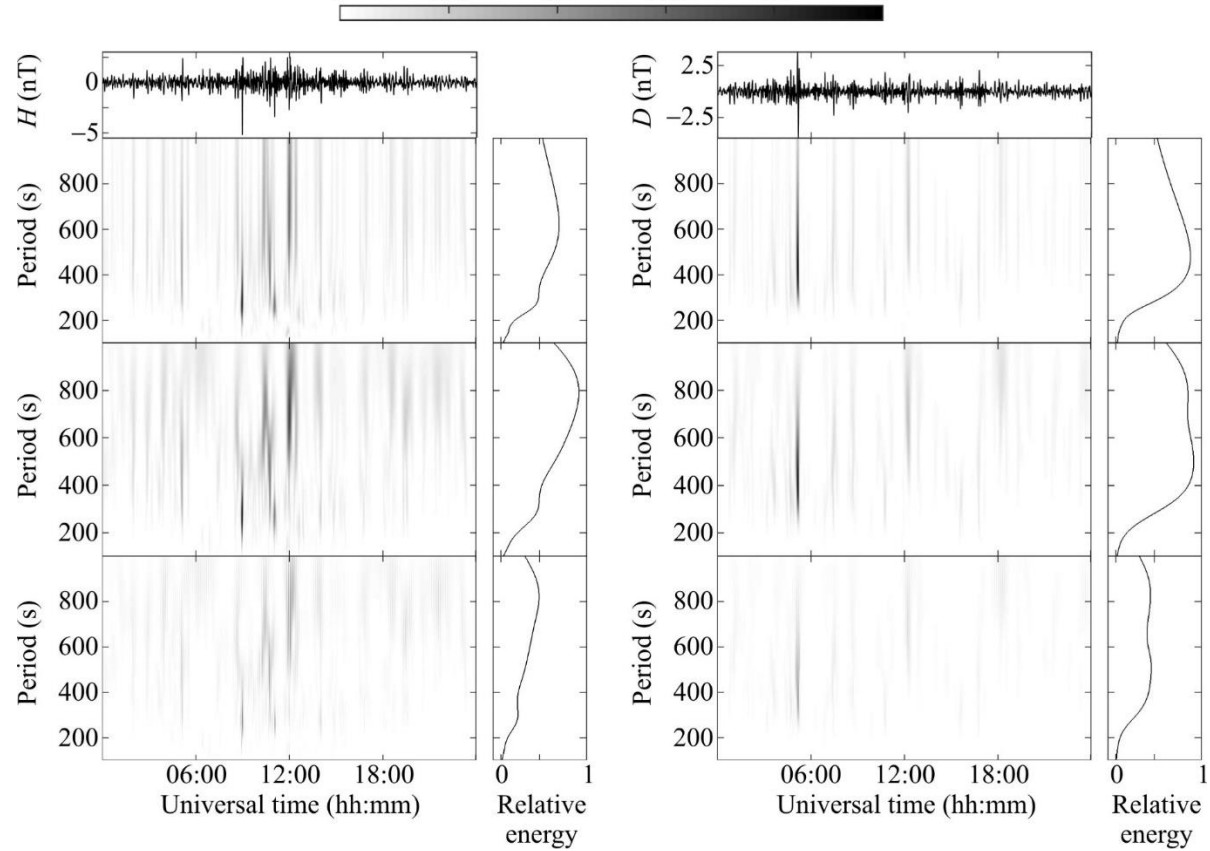


Figure 17: Systems spectral analysis products for the geomagnetic variations on 31 August 2019 at V. N. Karazin Kharkiv
National University Magnetometer Observatory.

Analysis of the ionograms indicates that the ionospheric storm occurred mainly during the 31 August 2019 and 1
September 2019 period. The storm duration virtually coincide with the duration of the magnetic storm main phase.
Since the $f_{oF2}$ values on 31 August 2019 were less than those on 29 August 2019, a reference day, by 0.7–1.1 MHz,
the ionospheric storm should be classified as negative. Furthermore, the $f_{oF2}$ values on 1 September 2019, were less than
those on 2 September 2019, another reference day.
Estimation of a decrease in the electron density, $N$, during the ionospheric storm as compared to the electron
density, $N_0$, on the reference day has been made using the following relation:
$$\frac{N_0}{N} = \left( \frac{f_{oF20}}{f_{oF2}} \right)^2 .$$

The dawn, daytime, and dusk $N_0/N$ ratio for 31 August 2019 were observed to be 1.8–2, 1.4, and 2.4, respectively.
The dawn and daytime $N_0/N$ ratio for 1 September 2019 was observed to be close to 1.56 and 1.16, respectively.
Given the $N_0/N$, the negative ionospheric index [Chernogor and Domnin, 2014] can be calculated
$$I_{NIS} = 10 \log_{10} \frac{N_0}{N_{min}} , \text{dB.}$$

For this storm, $(N_0/N_{min}) \approx 2.4$, and $I_{NIS} \approx 3.8$ dB. In accordance with Chernogor and Domnin's classification (2014), the
strength of such an ionospheric storm is classified as Negative Ionospheric Storm Index 3, NIS3. Furthermore, this geospace
storm manifested itself not only in the ionospheric $F$ region, but also in the ionospheric $E$ region, and in sporadic $E_s$ layer.
As a whole, the mechanisms for negative ionospheric storms are well known. They include an enhancement in the
wind speed, traveling atmospheric disturbances propagating equatorward (Prölss, 1993a, b), composition changes in the
thermosphere, and an increase from ~0.1—0.3 mV/m to 5—10 mV/m in an eastward zonal electric field arising during an
electrical storm (see, Section 1, Introduction) that acts to decrease the electron density and to increase $F_2$-layer virtual height.
The estimate of the ionospheric storm index and of the energy of the geospace and magnetic storms have allowed us
to establish that a weak geospace storm acted to give rise to a moderate magnetic storm and to a strong ionospheric storm,
which is not as trivial as may be supposed. The establishment of this fact were impossible without the quantitative estimates.
During ionospheric storms the phases/ionospheric response (positive and negative) are usually alternating. In most
cases, the CIR storms have positive effect just after storm onset. Storms are usually accompanied by large- or medium-scale
travelling ionospheric disturbances formed by GW that propagate from high latitudes toward the equator.
**7.3.2 Radio-wave reflection height variations**
The ionosonde data show that the virtual reflection heights $h'_E$, $h'_{Es}$, and $h'_{F2}$ exhibit sharp brief spikes at particular times.
This suggest significant changes occurring in the $N(h)$ profile. The variations in $N(h)$ acted to sharply change the Doppler
shift of frequency $f_D(t)$. On 31 August 2019, at about 14:00 UT, the $f_D$ virtually along all propagation paths exhibited a sharp
decrease from 0 Hz to –(1–1.5) Hz, followed by an increase from the minimum value to 0 Hz. This duration of this effect
was observed to be 50 to 60 min for different propagation paths. The sharp decrease in $f_D(t)$ followed by its increase to the
initial value indicates that a rise in the reflection height occurred. A rise in the altitude can be estimated by using the
following simplified relation:
$$\Delta z_r = -\frac{c}{4} \frac{\Delta f_{Dm}}{f} \left( \frac{\Delta T_1}{\cos \theta_1} + \frac{\Delta T - \Delta T_1}{\cos \theta_2} \right), \tag{1}$$

where c is the speed of light, $\Delta f_{Dm}$ is an $f_D$ maximum value, $\Delta T_1$ is the duration of a decrease in $f_D(t)$, $\Delta T$ is an overall
duration of the variation in $f_D$, $\overline{\cos \theta_1}$, and $\overline{\cos \theta_2}$ are values averaged over $\Delta T_1$ and $\Delta T - \Delta T_1$, respectively, and $\theta$ is an angle
of incidence with respect to the vertical.
Often, $\Delta T_1 = \Delta T - \Delta T_1$, i.e., $\Delta T_1 = \Delta T/2$. Hence, from Eq. (1), one has the relation
$$\Delta z_r = -\frac{c\Delta T}{4\cos\theta_{\mathrm{eff}}}\frac{\Delta f_{Dm}}{f},$$
where
$$\frac{1}{\cos\theta_{\mathrm{eff}}} = \frac{1}{2}\left(\frac{1}{\cos\theta_1}+\frac{1}{\cos\theta_2}\right).$$                 (2)
Then it follows from Eq. (1) and Eq. (2) that the altitude of reflection increases when $\Delta f_{Dm} < 0$, and vice versa.
The expression in Eq. (2), when applied to the Lintong/Pucheng–Harbin propagation path where $\Delta f_{Dm} \approx -1$ Hz and
$\Delta T = 60$ min for nighttime conditions, gives $\Delta z_r \approx 110$ km, i.e., the altitude exhibits an increase from ~150 km to ~260 km.
For the Hwaseong–Harbin propagation path, when $\Delta f_{Dm} \approx -1$ Hz and $\Delta T \approx 60$ min, the level of reflection shifts upward in
altitude from 150 km to 300–310 km. Regarding the mechanism for an increase in the height of reflection from 150 km to
300 km, such a large increase was observed at one time, 14:00 UT on August 31, 2019, when a few causes merged together.
First, the rearrangement of the evening ionosphere into the night ionosphere had been completed, which was accompanied
by a decrease in the electron density and an increase in the height of reflection. Second, due to the processes referred to
above, the negative ionospheric storm ensued. Third, a large negative half-wave of the quasi-periodic disturbance had
arrived, which was observed along all radio wave propagation paths from about 12:00 UT to 16:00 UT. Variations in the
height of reflection that occurred over other time intervals were observed to occur within the 30—50 km limits.
The altitudes of reflection along other propagation paths were estimated to be of the same order of magnitude. This effect is
also a manifestation of the ionospheric storm.
**7.3.3 Wavelike disturbance effects**
The ionospheric storm was accompanied by the generation of quasi-periodic variations in the Doppler shift of frequency.
From 12:00 UT to 17:00 UT on 31 August 2019, virtually all propagation paths exhibited a quasiperiodicity in $f_D(t)$ at ~30
min period, $T$, and ~0.4–0.6 Hz amplitude, $f_{Da}$. Given the $f_{Da}$, the amplitude of variations in the electron density can be
estimated by employing the following relation (Guo et al., 2019a, 2020; Chernogor et al., 2020):
$$\delta_{Na} = \frac{K}{4\pi}\frac{cT}{L}\frac{f_{Da}}{f},$$                 (3)
where $K = \dfrac{1+\sin\theta}{\left(1+2\zeta\tan^2\theta\right)\cos\theta}$ , $\zeta = \dfrac{z_r}{r_0}$ , $\tan\theta = \dfrac{R}{2z_r}$ , $L = \dfrac{2HL_n}{2H+L_n}$ , $z_r$ is the altitude of reflection, $r_0$ is the Earth's radius, $H$ is
the scale height of the atmosphere, $L_n$ is a characteristic scale length of changes in the refractive index in the ionosphere.
The expression in Eq. (3) suggests that
$$\delta_N(t,z) = \delta_{Na}(z_0)e^{(z-z_0)/2H}\cos\frac{2\pi t}{T},$$
where $z_0$ is a reference height, e.g., 100 km.
Applying the expression in Eq. (3) to, for example, the Hwaseong–Harbin propagation path, where $z_r \approx 150$ km, $f_{Da} = 0.4$ Hz,
$T = 30$ min, and $L \approx 30$ km, yields $\delta_{Na} \approx 42$ %. Along the Goyang–Harbin propagation path over the 17:30–20:00 UT period,
an oscillation with ~15 min period, $T$, and 0.1 Hz amplitude, $f_{Da}$, was observed to occur. Substituting $z_r \approx 200$ km and $L \approx 80$
km in Ed. (3) leads to $\delta_{Na} \approx 6$ %.
The magnitudes of periods, of ~15–60 min, and of the amplitudes $\delta_{Na}$ suggest that the quasi-periodic variations in
$f_D(t)$ and $N(t)$ launched atmospheric gravity waves (AGWs). It is well known that AGWs are generated in the auroral oval in
the course of geospace storms and propagate to low latitudes (see, for example, Hajkowicz, 1991; Lei et al., 2008; Lyons et
al., 2019). We have tried to find a confirmation of this fact in our measurements. For example, the minimum magnitude of
the Doppler shift of frequency along the Ulaanbaatar to Harbin (7,260 kHz) propagation path is observed to occur at

approximately 12:47 UT, and along the Beijing to Harbin (6,175 kHz) propagation path at 13:00 UT. Taking into account the distance of 400 km between the propagation path midpoints in the equatorward direction yields the equatorward speed of 510 m/s. Such speeds and periods of tens of minutes are inherent in atmospheric gravity waves. Thus, the generation of AGWs responsible for traveling ionospheric disturbances is also a manifestation of geospace storms.

### 7.3.4 Variations in radio wave characteristics

Ray tracing has shown that radio waves at frequencies equal to ~5–10 MHz were reflected from the ionosphere during the daytime at relatively low altitudes ($z \approx 100$–150 km) where the electron density was perturbed by the geospace storm relatively weakly, and the variations in $f_D$ usually did not exceed 0.1–0.2 Hz on both quiet and disturbed days. Under nighttime disturbed conditions, the altitude of reflection increased by 120–220 km, and the Doppler shift of frequency, $f_D$, exhibited significant aperiodic variations, from –1.5 Hz to +1.5 Hz and somewhat smaller (see Table 2). In contrast, during quiet days, such variations usually did not exceed ±(0.1–0.3) Hz. On 31 August 2019 and 1 September 2019, the quasi-periodic variations in the Doppler shift of frequency was observed to occur during night with amplitude, $f_{Da}$, of 0.2 Hz to 0.5 Hz and period of 24 min to 60 min (see Table 2), while the level of reflection oscillated with amplitude of ~10 km to ~20–30 km and traveled with velocity of ~10 m/s to ~60 m/s. Table 2 shows that amplitude variations on the disturbed day were considerably greater than the variations on the quiet day.

The studies presented at this paper demonstrate conclusively that the multi-frequency multipath facility involving the software-defined technology for sounding obliquely the ionosphere at the Harbin Engineering University is an effective means for investigating the influence of ionospheric storms on the characteristics of HF radio waves and the short-term variability of dynamic processes operating in the ionosphere.

### 8 Conclusions

1. The energy and power of the geospace storm have been estimated to be $1.5 \times 10^{15}$ J and $1.5 \times 10^{10}$ W, which means that this storm is classified as weak.

2. The energy and power of the magnetic storm have been estimated to be $1.5 \times 10^{15}$ J and $9 \times 10^{9}$ W, which means that this storm is classified as moderate. The storm's main feature is its main phase duration, of up to two days. The recovery phase was also long, no less than two days.

3. In the course of 31 August 2019 and 1 September 2019, the $H$- and $D$-component disturbances attained 60–70 nT. The $Z$-component variations did not exceed 20 nT.

4. On 31 August 2019 and 1 September 2019, the level of fluctuations in the geomagnetic field in the 1–1000 s period range exhibited an increase from 0.2–0.3 nT to 2–4 nT. The oscillations in the 300–400 s to 700–900 s period range had maximum energy.

5. During the geospace storm, a moderate to strong negative ionospheric storm was manifested by the reduction in the ionospheric $F$ region electron density on 31 August 2019 and 1 September 2019 by a factor of 1.4 to 2.4 times as compared to the values on the reference day.

6. In the course of the geospace storm, appreciable disturbances were observed to occur in the ionospheric $E$ region, and possibly in the $E_s$ layer.

7. The atmospheric gravity waves generated within the geospace storm period modulated the ionospheric electron density. The amplitude of the disturbances in the electron density could attain ~42 % at ~30 min period, while at ~15 min period it did not exceed 6 %.

Table 2
Aperiodic variations in the signal amplitude, $\delta A$, aperiodic and quasi-periodic variations in the Doppler shift of frequency, $f_D$,
with amplitude $f_{Da}$ and period $T$, as well as the amplitudes of variations in the level of reflection, $\Delta z_{ra}$, and in the speed of the
level of reflection, $v_a$.

| Radio station | Reference day (30 August 2019) | | Disturbed days (31 August 2019, 1 September 2019) | | | | | |
|---|---|---|---|---|---|---|---|---|
| | $f_D$ [Hz] | $\delta A$ [dBV] | $f_D$ [Hz] | $f_{Da}$ [Hz] | $\delta A$ [dBV] | $T$ [min] | $\Delta z_{ra}$ [km] | $v_a$ [m/s] |
| Lintong/Pucheng | 0.1–0.3 | 10 | (–1)– (+1.5) | 0.20– 0.25 | (15–20) | 40 | 18–22 | 46–58 |
| Hwaseong | ±0.4 | 10 | (–1)– (+0.6) | 0.4–0.7 | 30 | 40 | 13–24 | 35–62 |
| Chiba/Nagara | ±0.1 | 10 | (–1.4)– (+0.7) | 0.2–0.3 | 20 | 60 | 18–27 | 31–47 |
| Beijing (6,175 кГц) | 0–0.1 | 15 | (–0.7)– (+0.4) | 0.20 | 30 | 30 | 5.4 | 19 |
| Goyang | 0.1 | 15–20 | (–1.3)– (+0.7) | 0.2–0.5 | 30 | 40 | 6–14 | 15–38 |
| Ulaanbaatar | 0.1–0.3 | 5–10 | (–1.5)– (+1.0) | 0.10 | 30 | 20 | 2.3 | 12 |
| Yakutsk | 0.1 | 5–10 | (–1.2)– (+1.5) | 0.2 | 10–20 | 24 | 7.2 | 31 |
| Shijiazhuang | 0.1 | 10–15 | (–1)–(+0.8) | 0.3–0.5 | 20 | 30 | 4.7–4.8 | 16–27 |
| Hohhot | 0.1–0.2 | 10 | (–0.5)– (+0.7) | 0.1–0.5 | 20–40 | 24 | 1.2–6.2 | 5–27 |
| Yamata | 0.1–0.2 | 10–15 | (–0.5)– (+1.0) | 0.1–0.3 | 40–60 | 20–30 | 2–6 | 8–24 |
| Beijing (9,830 kHz) | 0–0.1 | 10–20 | (–0.3)– (+1.0) | 0.2–0.5 | 20–30 | 20–30 | 2–5 | 8–20 |


8. In the course of the ionospheric storm, the Doppler shift of frequency could show a sharp decrease to –1.5 Hz or increase
to +1.5 Hz while the height of reflection could exhibit a sharp increase from ~150 km to ~300 – 310 km and then a decrease
of the same magnitude. On quiet days, the variations in the Doppler shift of frequency usually do not exceed ±(0.1–0.2) Hz.
9. The quasi-periodic disturbances in the electron density acted to periodically move the level of reflection of radio waves
with ~10–60 m/s speed and an oscillation amplitude attaining ~20–30 km.
10. The variations in the signal amplitude attained 30–60 dBV during the ionospheric storm, while on quiet days they did not
exceed 15–20 dBV.
11. The ionospheric storm effects manifest themselves more distinctively under nighttime conditions when the radio waves
are reflected from the more disturbed ionospheric $F$ region.
12. The ionospheric HF radio channel is substantially affected by both the moderate and strong ionospheric storms.

**Code availability**
The doppler14.grc file contains the computer program code that generates the data from the raw data recorded by the multi-
frequency multipath system at the Harbin Engineering University campus, the People's Republic of China (45.78° N,
126.68° E). These data are needed to plot the Doppler shift of frequency and the amplitude presented in Figures 5–15 (see
the SupplementaryMaterial.zip file).

**Data availability**
The raw data sets recorded by the multi-frequency multipath system at the Harbin Engineering University campus, the
People's Republic of China (45.78° N, 126.68° E) and discussed in this paper can be requested online at
https://dataverse.harvard.edu/dataset.xhtml?persistentId=doi:10.7910/DVN/86LHDC (Luo et al., 2020b).
Citation:
Luo, Y., Chernogor, L., Garmash, K., Guo, Q., Rozumenko, V., Zheng, Y.: RAW Data on Parameters of Ionospheric HF
Radio Waves Propagated Over China During the August 30–September 2, 2019 Geospace Storm,
https://dataverse.harvard.edu/dataset.xhtml?persistentId=doi:10.7910/DVN/86LHDC, 2020b.

**Author contribution**. https://casrai.org/credit/
Yiyang Luo processed the data observed. Leonid Chernogor interpreted the physics of the observations, wrote Sections 1, 6,
7, and 8. Kostiantyn Garmash developed the software, processed the data, wrote Sub-sections 2.1. Qiang Guo developed the
software, conducted uninterrupted observations. Victor Rozumenko wrote Section 3, and Subsections 4.1, 4.2, 5.2. Yu Zheng
wrote Subsections 2.2, 5.1. All co-authors took part in the discussion of the results obtained.

**Competing interests**
The authors declare that they have no conflict of interest.

**Acknowledgments**. The solar wind parameters have been retrieved from the Goddard Space Flight Center Space Physics
Data Facility https://omniweb.gsfc.nasa.gov/form/dx1.html. This publication makes use of the data recorded at the Low
Frequency Observatory, owned by the Institute of Radio Astronomy NASU, Radiophysics of Geospace Department,
Laboratory of Electromagnetic Surrounding of the Earth. The authors thank the staff of the Observatory for its operation
(magnetometer data are retrieved from http://geospace.com.ua/en/observatory/metmag.html). This research also draws upon
data provided by the WK546 URSI code ionosonde at the City of Wakkanai (45.16° N, 141.75° E), Japan, URL:
wdc.nict.go.jp/IONO/HP2009/contents/Ionosonde_Map_E.html. Ionosonde data from the City of Moscow, the Russian
Federation (55.47° N, 37.3° E), are retrieved from
https://lgdc.uml.edu/common/DIDBYearListForStation?ursiCode=MO155. Graduate student Luo Yiyang thanks the China
Scholarship Council (CSC) program (201908100008) for the financial support. Work by Qiang Guo and Yu Zheng was
supported by National key R & D plan strategic international science and technology cooperation and innovation
(2018YFE0206500). Work by L. F. Chernogor and Y. Luo was supported by the National Research Foundation of Ukraine
for financial support (project 2020.02/0015, "Theoretical and experimental studies of global disturbances from natural and
technogenic sources in the Earth-atmosphere-ionosphere system"). Work by L. F. Chernogor, K. P. Garmash, and by V. T.
Rozumenko was supported by Ukraine state research project #0119U002538.

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
