# Peer review of "Dynamic processes in the magnetic field and in the ionosphere"

_Annales Geophysicae, 2020_

## Referee Comment (RC1) · Anonymous Referee #1 · 11 Sep 2020

The paper contains a large volume of information on the behavior of the ionophere during the magnetic storm of August 31-September 01, 2019. The main part of that information is obtained by the oblique sounding observations at 10 ionospheric paths with the receiver in Harbin (China). Quite simlar study of the Aygust 26, 2018 magnetic storm is published by the same group of authors in Geomagnetism and Aeronomy 60 (6), 2020.

As far as studying ionospheric reaction to magnetic storms is of a great importance for many applied problems and very complicated due to a different reaction of the ionosphere in different conditions, I thnk that any information on the ionosperic behavior

during a particular storm is important and deserves publication.

The authots provide a valuable information on the behavior of the ionospheric parameters at 10 paths. They also describe in detail the dehavior of the ionospheric layers during the storm days as compared to the quiet days according to the vertical soundind data by the ionosondes in Japan and Moscow.

I think that the aforementioned results of observations present the most valuable part of the paper. There are also some conclusions of the authors on the physical processes in the upper atmosphere and ionosphere (for example, on generation of IGW ) however they could be considered only as an example of using the observed data for aeronomical considerations and could be doubled.

Thus, I think that the paper should be published in AG. However, in my mind, is needs some minor corrections.

The statement in the abstract that "...L.F. Chernogor validated that the concept of geospace storms..." is not modest and is incorrect. The studies of complex processes that occur in the terrestrial environment in disturbed solar conditions have benn published much earlier by several authors (for example, H. Rishbeth, G. Prollss and others). I recommend to withdrawn this sentence from the abstract.

The term geospace is not commonly used. What is actually described in the paper, is the ionospheric reaction to the particulat magnetic storm. And that is how the situation is described in many studies of that kind. The source of the geomagnetic disturbance is not important in that case.

The references to main previous publications is not complete enough. In the aforementioned publications on the 2018 storm, the reference is more complete.

---

## Author Comment (AC1) · 18 Sep 2020

Dear Anonymous Referee #1,

Thank you very much for your valuable comments.

Anonymous Referee #1 Comment #1:

The statement in the abstract that "...L.F. Chernogor validated that the concept of geospace storms..." is not modest and is incorrect. The studies of complex processes that occur in the terrestrial environment in disturbed solar conditions have been published much earlier by several authors (for example, H. Rishbeth, G. Prollss and others). I recommend to withdrawn this sentence from the abstract. Authors' response to Anonymous Referee #1 Comment #1:

The authors have corrected the statement concerning the validation of the concept of geospace storms. We have modified the sentence as follows:

The concept that geospace storms are comprised of synergistically coupled magnetic storms, ionospheric storms, atmospheric storms, and storms in the electric field originating in the magnetosphere, the ionosphere and the atmosphere (i.e., electric storms) has been validated a few decades ago.

These changes in the manuscript are marked in red.

As far as we know, H. Rishbeth and G. Prollss consider magnetic storms and ionospheric storms separately, and they do not even mention electric storms.

Anonymous Referee #1 Comment #2:

The term geospace is not commonly used. What is actually described in the paper, is the ionospheric reaction to the particulat magnetic storm. And that is how the situation is described in many studies of that kind. The source of the geomagnetic disturbance is not important in that case.

Authors' response to Anonymous Referee #1 Comment #2:

The term geospace was brought into usage a long time ago, and it is commonly used now. This statement is supported by the following haphazard review of research in the field:

When one visits the MIT Haystack Observatory website https://www.haystack.mit.edu/ today, he or she can see the Geospace Sciences web page https://www.haystack.mit.edu/geospace//

Also, at present, the National Oceanic and Atmospheric Administration (NOAA) Space Weather Prediction Center at (https://www.swpc.noaa.gov/products/geospacemagnetosphere-movies) provides Geospace Magnetosphere Movies from the University of Michigan Geospace model output.

Back in July 2013, the Journal of Atmospheric and Solar-Terrestrial Physics published a special issue (Volume 99, pages 1–164) Dynamics of the Complex Geospace System Edited by Vania K. Jordanova, Joseph E. Borovsky, Ilia Roussev. https://www.sciencedirect.com/journal/journal-of-atmospheric-and-solar-terrestrial-physics/vol/99/suppl/C

The book Extreme Events in Geospace: Origins, Predictability, and Consequences written by M. Hapgood, N. Gopalswamy, K.D. Leka, G. Barnes, Yu. Yermolaev, P. Riley, S. Sharma, G. Lakhina, B. Tsurutani, C. Ngwira, A. Pulkkinen, J. Love, P. Bedrosian, N. Buzulukova, M. Sitnov, W. Denig, M. Panasyuk, R. Hajra, D. Ferguson, S. Lai, L. Narici, K. Tobiska, G. Gapirov, A. Mannucci, T. Fuller-Rowell, X. Yue, G. Crowley, R. Redmon, V. Airapetian, D. Boteler, M. MacAlester, S. Worman, D. Neudegg, and M. Ishii and edited by Natalia Buzulukova is concerned with Geospace. https://www.elsevier.com/books/extreme-events-in-geospace/buzulukova/978-0-12-812700-1

As is well known, the term geospace include the magnetosphere (geomagnetic field), the ionosphere, and the upper atmosphere. Our paper is concerned with the effects in the magnetic field, the ionosphere, and the effects in the upper atmosphere are only mentioned. Therefore, we think that the term geospace is appropriately used in this paper.

Anonymous Referee #1 Comment #3:

The references to main previous publications is not complete enough. In the aforementioned publications on the 2018 storm, the reference is more complete

Authors' response to Anonymous Referee #1 Comment #3:

Indeed, some key earlier papers are missed. Thank you very much for this comment.

We have added references to the following three publications:

1. Prölss G.W. Magnetic storm associated perturbations of the upper atmosphere / Magnetic storms. Eds. Tsurutani B.T., Gonzalez W.D., Kamide Y., Arballo J.K. // Geoph. Monog. Series. V. 98. Washington, D.C.: AGU. P. 249–290. 1997. https://doi.org/10.1029/GM098p0227

2. Danilov, A. D. and Morozova, L. D., 1985. Ionospheric storms in the F2 region. Morphology and physics (review) (in Russian). Geomagnetism and Aeronomy. Vol. 25, no. 5, pp. 705–721.

3. Danilov, A. D. Reaction of F region to geomagnetic disturbances (Review) (in Russian), Heliogeophysical Research, No. 5, pp. 1–33, 2013. https://www.elibrary.ru/item.asp?id=21273665

These additions in the manuscript are marked in red.

However, the reader would find Paper 2 above difficult to access via the World Wide Web, because it is written in 1985.

---

## Referee Comment (RC2) · Anonymous Referee #2 · 20 Dec 2020

The paper by Luo et al. investigated the storm energy input and ionospheric storm effects during a storm event. They estimated the energy, power, magnetic field variation, F-region density decrease, E-region lifting, and density modulation. They interpreted that the density modulation is cause by atmospheric gravity waves.

The paper presents interesting data but gives very little data interpretation. It is difficult for readers to find out what physical mechanisms contributed to create the ionospheric storm. Please extend the discussion of the key observation results of the ionospheric storm. The storm energy and magnetic field variation have already been studied in earlier works and I don't think that the present paper gives any new aspects of them.

[Figure]

I suggest to remove them from the conclusion and focus on the findings on the ionospheric storm.

Line 15-23 Basic properties of storms (energy, power, duration and magnetic field variations) are already documented more in details in existing papers (e.g., Gonzalez et al., 1994, Knipp et al., 1998, Feldstein et al., 2003). I don't think these are not unique features. These sentences should be removed and the abstract should focus on the ionospheric storm part.

For the same reason, the title of the paper is not very informative. I suggest to remove the magnetic field and instead specify what dynamic processes are the major finding of this paper.

The paper uses the network of radio wave observations but does not discuss how the ionospheric storm effects vary in latitude or longitude. What spatial dependence of the ionospheric storm effects do the authors see? Are those localized in specific areas or do they propagate in any direction? Such information will be useful to interpret the mechanism of the ionospheric storm.

What is the cause of the negative ionospheric storm? Is the F-region density decrease created by vertical motion of the ionosphere, horizontal motion, or change in thermospheric composition?

What is the interpretation of the driver of the increased height of the reflection from 150 to 300 km? Is it because of the penetration electric field?

Have the authors looked at the total electron content data from GNSS receivers for this storm? The total electron content covers much wider areas and could be useful to show how the distribution of the ionosphere density changed and what density structure gave the negative storm.

The authors suggest that the quasi-periodic variations are caused by atmospheric gravity waves generated in the auroral oval. In the radio wave data, do you see that the

quasi-periodic variations propagate from higher to lower latitudes? If the equatorward propagation is seen, it could be a supportive evidence of the auroral source.

---

## Author Comment (AC2) · 6 Jan 2021

Dear Anonymous Referee #2,

Thank you very much for a comprehensive study of the text and your valuable recommendations.

With regard to your comments, we can reply the following.

Anonymous Referee #2 Comment #1: The paper by Luo et al. investigated the storm energy input and ionospheric storm effects during a storm event. They estimated the energy, power, magnetic field variation, F-region density decrease, E-region lifting,

and density modulation. They interpreted that the density modulation is cause by atmospheric gravity waves.

Authors' response to Anonymous Referee #2 Comment #1: Indeed, the paper deals with a storm event.

Anonymous Referee #2 Comment #2: The paper presents interesting data but gives very little data interpretation. It is difficult for readers to find out what physical mechanisms contributed to create the ionospheric storm. Please extend the discussion of the key observation results of the ionospheric storm. The storm energy and magnetic field variation have already been studied in earlier works and I don't think that the present paper gives any new aspects of them. I suggest to remove them from the conclusion and focus on the findings on the ionospheric storm.

Authors' response to Anonymous Referee #2 Comment #2: The information on the magnetic storm and its energy should not be removed from the paper for the following reasons. (1) In accordance with the aim of the study, the analysis has been undertaken of the dynamical processes that occurred in the course of the geospace storm, which is comprised of the magnetic, ionospheric, atmospheric, and electrical storms. (2) The magnetic storm in question was not yet described by anybody except for the authors of this paper. (3) Any two magnetic storms are not identical, and the estimation of the basic parameters of each new storm will ever remain a problem of current importance. For example, the basic parameters of every new earthquake, are also described for exactly the same reason. (See also the (4th) point in Authors' response to Anonymous Referee #2 Comment #3)

Anonymous Referee #2 Comment #3: Line 15-23 Basic properties of storms (energy, power, duration and magnetic field variations) are already documented more in details in existing papers (e.g., Gonzalez et al., 1994, Knipp et al., 1998, Feldstein et al., 2003). I don't think these are not unique features. These sentences should be removed and the abstract should focus on the ionospheric storm part.

[Figure]

5Authors' response to Anonymous Referee #2 Comment #3: Gonzalez et al. (1994) made an excellent review summarizing information on geomagnetic storms up to the early 1990s. From that time on, significant progress in understanding geomagnetic storms has been made. The authors have used a relation from the paper by Gonzalez et al. (1994) for estimating the magnetic storm energy and the power (see Section 7.1 Geomagnetic field effects, Line 404). Knipp and Emery (1998) described in detail the processes accompanying the November 2–11, 1993 geomagnetic storm. Feldstein et al. (2003) analyzed in detail the energy of the processes acting in the magnetosphere during two particular storms. In the Introduction, the following text has been added (Line 63–66):

In particular, Gonzalez et al. (1994) made an excellent review summarizing information on geomagnetic storms up to the early 1990s. Knipp and Emery (1998) described in detail the processes accompanying the November 2–11, 1993 geomagnetic storm. Feldstein et al. (2003) analyzed in detail the energy of the processes acting in the magnetosphere during two particular storms. In the Reference list, the following two references has been added: Feldstein, Y.I., Dremukhina, L.A., Levitin, A.E., Mall, U., Alexeev, I.I., and Kalegaev, V.V.: Energetics of the magnetosphere during the magnetic storm, J. Atmos. Terr. Phys., 65 (4), 429–446, ISSN 1364-6826, https://doi.org/10.1016/S1364-6826(02)00339-5, 2003.

Knipp, D.J. and Emery, B.A.: A report on the community study of the early November 1993 geomagnetic storm, Advances in Space Research, 22 (1), 41-54, , DOI: 10.1016/S0273-1177(97)01098-3,1998.

Since energy characteristics for various geomagnetic storms vary by orders of magnitude, the estimates of these characteristics should be made for each new storm. We have estimated the energy characteristics of the geospace storm, as well as of its constituent parts viz the magnetic storm and the ionospheric storm. (4) The energy estimates of geospace and magnetic storms, and the ionospheric storm index permitted us to establish the non-trivial fact that a weak geospace storm acted to give rise

to a moderate magnetic storm and to a strong ionospheric storm. The establishment of this fact were impossible without the quantitative estimates. This fact has been described in the paragraph that has been added to the text at the end of Section 7.3.1 (Ionogram Parameter Disturbances): The estimate of the ionospheric storm index and of the energy of the geospace and magnetic storms have allowed us to establish that a weak geospace storm acted to give rise to a moderate magnetic storm and to a strong ionospheric storm, which is not as trivial as may be supposed. The establishment of this fact were impossible without the quantitative estimates.

Anonymous Referee #2 Comment #4: For the same reason, the title of the paper is not very informative. I suggest to remove the magnetic field and instead specify what dynamic processes are the major finding of this paper.

Authors' response to Anonymous Referee #2 Comment #4: It does not seem to us that the paper title is not very informative. The paper deals with the manifestations of the magnetic and ionospheric storms that are integral parts of the geospace storm. The manifestations of the atmospheric and electrical storms are observed in the indirect features: (1) in quasi-periodic variations in the Doppler shift of frequency under the action of atmospheric gravity waves, and (2) in the vertical plasma drifts in the ionosphere under the action of the enhanced zonal electric field.

Anonymous Referee #2 Comment #5: The paper uses the network of radio wave observations but does not discuss how the ionospheric storm effects vary in latitude or longitude. What spatial dependence of the ionospheric storm effects do the authors see? Are those localized in specific areas or do they propagate in any direction? Such information will be useful to interpret the mechanism of the ionospheric storm.

Authors' response to Anonymous Referee #2 Comment #5: This storm, as any other storm, was global in nature. At least it occupied all area of the People's Republic of China. The global character of the storm is also supported by the magnetic measurements at the City of Kharkiv, Ukraine, and by the state of the ionosphere as inferred

from the ionosonde data at the City of Moscow, the Russian Federation, and at the City of Wakkanai, Japan. The investigation of longitudinal and latitudinal dependences was not a purpose of the paper (see line 12–15). Such an investigation requires measurements different from those used in this study.

Anonymous Referee #2 Comment #6: What is the cause of the negative ionospheric storm? Is the F-region density decrease created by vertical motion of the ionosphere, horizontal motion, or change in thermospheric composition?

Authors' response to Anonymous Referee #2 Comment #6: The authors have not intended to discuss mechanisms for the negative storm (see line 12–15). The subject of this study is the influence of the ionospheric storm on the characteristics of HF radio waves propagating over the area of the People's Republic of China. As a whole, the mechanisms for negative ionospheric storms are well known (e.g., see Section 11.16, Ionospheric storms, in the textbook of R.W. Schunk and A. F. Nagy Ionospheres: Physics, plasma physics, and Chemistry, Cambridge University Press, 2009, ISBN-13 978-0-521-87706-0). They include an enhancement in the wind speed, traveling atmospheric disturbances propagating equatorward [Prölss G. W. On explaining the local time variation of ionospheric storm effects, Ann. Geophys. 1993. Vol. 11. pages 1–9; Prölss G. W. Common Origin of Positive Ionospheric Storms at Middle Latitudes and the Geomagnetic Activity Effect at Low Latitudes, J. Geophys. Res. 1993. Vol. 98, No. A4. pages 5981−5991], composition changes in the thermosphere, and an increase from ∼0.1—0.3 mV/m to 5—10 mV/m in an eastward zonal electric field arising during an electrical storm (see, Section 1, Introduction) that acts to decrease the electron density and to increase F2-layer virtual height.

In the paper, the following text is added (at the end of Section 7.3.1 (Ionogram Parameter Disturbances)): As a whole, the mechanisms for negative ionospheric storms are well known. They include an enhancement in the wind speed, traveling atmospheric disturbances propagating equatorward (Prölss, 1993a, b), composition changes in the thermosphere, and an increase from ∼0.1—0.3 mV/m to 5—10 mV/m in an east-

ward zonal electric field arising during an electrical storm (see, Section 1, Introduction) that acts to decrease the electron density and to increase F2-layer virtual height.

In the Reference list, the following two references has been added: Prölss, G. W.: On explaining the local time variation of ionospheric storm effects, Ann. Geophys., 11 (1), 1–9, 1993.

Prölss, G. W.: Common Origin of Positive Ionospheric Storms at Middle Latitudes and the Geomagnetic Activity Effect at Low Latitudes, J. Geophys. Res., 98 (A4), 5981−5991, https://doi.org/10.1029/92JA02777, 1993.

Anonymous Referee #2 Comment #7: What is the interpretation of the driver of the increased height of the reflection from 150 to 300 km? Is it because of the penetration electric field?

Authors' response to Anonymous Referee #2 Comment #7: Regarding the increased height of the reflection from 150 to 300 km, such a large increase was observed one time at 14:00 UT on August 31, 2019, when a few reasons had been added together. First, the rearrangement of the evening ionosphere into the night ionosphere had been completed, which was accompanied by a decrease in the electron density and an increase in the height of reflection. Second, due to the processes referred to above, the negative ionospheric storm ensued. Third, a large negative half-wave of the quasi-periodic disturbance had arrived, which was observed along all radio wave propagation paths from about 12:00 UT to 16:00 UT. Variations in the height of reflection that occurred over other time intervals were observed to occur within the 30—50 km limits. In the text of the paper, this information has been added to the paper as follows. Regarding the mechanism for an increase in the height of reflection from 150 km to 300 km, such a large increase was observed at one time, 14:00 UT on August 31, 2019, when a few causes merged together. First, the rearrangement of the evening ionosphere into the night ionosphere had been completed, which was accompanied by a decrease in the electron density and an increase in the height of reflection. Second, due to the processes referred to above, the negative ionospheric storm ensued. Third, a large negative half-wave of the quasi-periodic disturbance had arrived, which was observed along all radio wave propagation paths from about 12:00 UT to 16:00 UT. Variations in the height of reflection that occurred over other time intervals were observed to occur within the 30–50 km limits.

Anonymous Referee #2 Comment #8: Have the authors looked at the total electron content data from GNSS receivers for this storm? The total electron content covers much wider areas and could be useful to show how the distribution of the ionosphere density changed and what density structure gave the negative storm.

Authors' response to Anonymous Referee #2 Comment #8: The authors did not studied variations in the total electron content. Such an investigation could be the subject of a separate paper. It has not been a purpose of this paper (see line 12–15).

Anonymous Referee #2 Comment #9: The authors suggest that the quasi-periodic variations are caused by atmospheric gravity waves generated in the auroral oval. In the radio wave data, do you see that the quasi-periodic variations propagate from higher to lower latitudes? If the equatorward propagation is seen, it could be a supportive evidence of the auroral source.

Authors' response to Anonymous Referee #2 Comment #9: It is well known that geospace storms are accompanied by the generation of atmospheric gravity waves in the auroral zone [R.W. Schunk and A. F. Nagy Ionospheres: Physics, plasma physics, and Chemistry, Cambridge University Press, 2009 (ISBN-13 978-0-521-87706-0)]. In the paper, as an example, three references are made (Hajkowicz, 1991; Lei et al., 2008; Lyons et al., 2019). We have tried to find a confirmation of this fact in our measurements. For example, the minimum magnitude of the Doppler shift of frequency along the Ulaanbaatar to Harbin (7,260 kHz) propagation path is observed to occur at approximately 12:47 UT, and along the Beijing to Harbin (6,175 kHz) propagation path at 13:00 UT. Taking into account the distance of 400 km between the propagation path midpoints in the equatorward direction yields the equatorward speed of 510 m/s. Such speeds and periods of tens of minutes are inherent in atmospheric gravity waves. In the text of the paper, this information has been added to the paper as follows. We have tried to find a confirmation of this fact in our measurements. For example, the minimum magnitude of the Doppler shift of frequency along the Ulaanbaatar to Harbin (7,260 kHz) propagation path is observed to occur at approximately 12:47 UT, and along the Beijing to Harbin (6,175 kHz) propagation path at 13:00 UT. Taking into account the distance of 400 km between the propagation path midpoints in the equatorward direction yields the equatorward speed of 510 m/s. Such speeds and periods of tens of minutes are inherent in atmospheric gravity waves.

The authors have tried to give answers to all Referee #2's comments. Sincerely, Authors.

Please also note the supplement to this comment:
https://angeo.copernicus.org/preprints/angeo-2020-57/angeo-2020-57-AC2-supplement.pdf

---

## Author Response (AR1)

Dear Dalia Buresova,

The Authors are grateful for the valuable recommendations, which have allowed the Authors to greatly improve the quality of the manuscript.

**A list of changes and a rebuttal against each point, which has been raised in your Comments to the Authors.**

**(1) Comment to the Authors:**
Let's start from the title. One of the referees suggested modification of the title and I fully agree with his opinion. In your response to the referee's comments you stated that " The subject of this study is the influence of the ionospheric storm on the characteristics of HF radio waves propagating over the area of the People's Republic of China". The title of the manuscript should therefore reflect the aim and substance of the research carried out.
**Authors' reply:**
The Authors agree to specify the paper's title as follows: «Influence of 31 August – 1 September, 2019 Ionospheric Storm on HF Radio Wave Propagation».

**(2) Comment to the Authors:**
You have introduced a large number of values of different solar wind, geomagnetic and ionospheric parameters, nevertheless they are not fully used for interpretation of your findings. So, my suggestion is to change the title of the manuscript and focus on the main goal "influence of ionospheric storm on HF radio wave propagation" instead of widely describing geospace storms.
**Authors' reply:**
The description of the geomagnetic storm is a must, for the geomagnetic storm is one of the constituents of the geospace storm. This is the essence of the systems approach, the essence of the systems paradigm (see, e.g., [Chernogor, Rozumenko, 2008]).

**(3) Comment to the Authors:**
On the other hand, the space weather event you are analysing in the paper, is not a CME-related event, but the event of CIR/CH HS origin combined with solar sector boundary crossing event, which could also affect geomagnetic situation. The NOAA informed that " from midday on 30 Aug through 01 Sep, field activity increased to unsettled to G1 (minor) and G2 (moderate) levels as Earth came under the influence of a large, recurrent positive polarity CH HSS. 30 Aug saw a SSBC from a negative to a positive sector in advance of a CIR, all preceding the CH HSS. 31 Aug and 01 Sep observed active to G1 and G2 storm conditions. Wind speeds averaged about 750 km/s during this time frame with a peak of 835 km/s observed early on 01 Sep" (SWPC PRF 2296 02 September 2019). These events have a bit different characteristics and courses compared with those of the CME-origin, e.g. smaller effect on Dst index, not so well expressed single phases of the storm, longer entire duration of the disturbances and, in total, larger energy input into the Earth's upper atmosphere (for more information, please, see Koskinen HEJ, (2011). Physics of space storms. From Solar Surface to the Earth, Springer in association with Praxis Publishing. DOI:10.1007/978-3-642-00319-6.
**Authors' reply:**
First, the Authors nowhere assert that the storm under consideration has been caused by a coronal mass ejection. Second, to avoid the misunderstanding, at the very beginning of 3 Analysis of the space weather state Section, and 6 lines below, the Authors have added the following two pieces of information (marked in magenta …) kindly suggested by Topical Editor:
The space weather variations under study are the event of CIR/CH HS origin combined with solar sector boundary crossing event, which could affect geomagnetic situation (see ftp://ftp.swpc.noaa.gov/pub/warehouse/2019/WeeklyPDF/prf2296.pdf; Koskinen, 2011).

and

"After 12:00 UT on 30 August 2019 through about 01:00 UT on 1 September 2019, the $V_{sw}$ value exhibited an increase from ~400 km s$^{-1}$ to 750 km s$^{-1}$ with a peak of 835 km/s observed early on 1 September 2019 (see ftp://ftp.swpc.noaa.gov/pub/warehouse/2019/WeeklyPDF/prf2296.pdf).

The reference to the monograph by Koskinen, 2011, has been added to the list of reference as follows: Koskinen, H.E.J.: Physics of space storms. From Solar Surface to the Earth, Springer in association with Praxis Publishing, DOI:10.1007/978-3-642-00319-6. 2011.

**(4) Comment to the Authors:**
During ionospheric storms the phases/ionospheric response (positive and negative) are usually alternating. In most cases the CIR storms have positive effect just after storm onset. Storms are usually accompanied by large- or medium-scale travelling ionospheric disturbances formed by GW that propagate from high latitudes toward the equator.
**Authors' reply:**
4. Taking into account the Topical Editor's recommendation, the Authors have added the following piece of information kindly suggested by Topical Editor (at the end of 7.3.1 Disturbances in ionogram parameters Section):
During ionospheric storms the phases/ionospheric response (positive and negative) are usually alternating. In most cases, the CIR storms have positive effect just after storm onset. Storms are usually accompanied by large- or medium-scale travelling ionospheric disturbances formed by GW that propagate from high latitudes toward the equator.

**(5) Comment to the Authors:**
My suggestion is to compare the courses of ionospheric parameters with monthly median values or running means (or compare with the courses for quiet days near the even) to see real response to the storm-induced disturbances.
**Authors' reply:**
The ionosonde data are used to characterize the general state of the ionosphere only, as it has been done in the "5.1 Data from ionosonde in Japan" and "Data from ionosonde at Moscow" Sections. Also, the comparison with monthly median values or running means does not contribute to understanding the problem of the influence of the ionospheric storm on HF radio wave propagation. Therefore, we have decided to remove Figure 5 from the paper.

**(6) Comment to the Authors:**
Now some minor comments. It seems that the ionospheric data you are using have some gaps, e.g. Fig.5, foF2 and h'(F2) at bout 05-08 am and 11am -15 pm. Please, don't use a solid line to cross the data gaps. Are your ionospheric data manually checked or are coming from automatic scaling?
**Authors' reply:**
The ionospheric data have been checked manually. The gaps are due to the screening by the $E_s$ layer (not depicted in Figure 5). With the solid lines removed, Figure 5 is shown below at the end of this Authors' reply (it has not been included in the paper). It could be seen that the data for the $F_2$ layer on 30 August 2019 are especially sparse, the gaps amount to about 30 percent of the time.

**(7) Comment to the Authors:**
"The manifestations of geospace storms vary over the solar cycle, and depend on season, local time, latitude, longitude, and observational facilities" (between lines 65-70). Are you really sure that manifestation of storms depends on observation facilities?
**Authors' reply:**
The phrase "manifestation of storms depends on … observation facilities" should be understood in such a way that each technique can determine disturbance in the medium parameter, which the observation facility is designed to determine. To exclude misunderstanding, we have replaced the words "observation facilities" with the words "… and so on".

**(8) Comment to the Authors:**

"The main feature of this geospace storm is its duration, of up to four days."(between lines 70-75). The duration of storms could be quite different depending on driver (from several hours up to 8-10 days).

**Authors' reply:**

We have made a more accurate assertion: "One of the interesting features of this geospace storm is its duration, of up to four days."

**(9) Comment to the Authors:**

"Thus, this magnetic storm had the longest duration observed over the last few years,…" (part 135-140). This statements is also doubtful.

**Authors' reply:**

We have made a more accurate assertion: "Thus, this magnetic storm was seen to be of quite a long duration over the last few years, …".

**(10) Comment to the Authors:**

"To assess the ionospheric storm on the global scale, ionosonde data from the City of Moscow (55.47°N, 37.30°E), the Russian Federation, have been used." (between 170-175). I guess that expression "global" usually is used for description of the phenomenon around all the globe and both hemispheres.

**Authors' reply:**

We have replace the term «global» with "characteristic extent of"(line 106, 180 ) and with «large-scale» (line 437).

**(11) Comment to the Authors:**

"…subsequently, a decrease from 15 x106 m–3 to to ? 106 m–3 in the course of the next three days" (between 115-120).

**Authors' reply:**

We have amended this phrase as follows: subsequently, a decrease from $15 \times 10^6$ m$^{-3}$ to $1\times10^6$ m$^{-3}$ in the course of the next three days

**(12) Comment to the Authors:**

If you are prepared to undertake the improvements required, please submit the revised manuscript as well as a list of changes or a rebuttal against each point, which is being raised when you submit the revised manuscript.

**Authors' reply:**

The authors have tried to give answers to all Topical Editor's comments.

Sincerely,
Authors.

[Figure]

Figure 5 with the solid lines removed. The ionospheric data have been checked manually. The gaps are due to the screening by the $E_s$ layer (not depicted in Figure 5). This figure has not been included in the paper.

---

## Author Response (AR2)

Dear Dalia Burešová,

The Authors are grateful for the valuable recommendations, which have allowed the Authors to greatly improve the quality of the manuscript.

All changes in the manuscript are marked in ==yellow==.

As time goes by, new interesting results are being published. Thus, the Authors have enlarged the review part of the paper and have added the reference to the paper [==Habarulema, J. B., Katamzi-Joseph, Z. T., Burešová, D., Nndanganeni, R., Matamba, T., Tshisaphungo, M., Buchert, S., Kosch, M., Lotz, S., Cilliers, P., and Mahrous, A. (2020). Ionospheric response at conjugate locations during the 7–8 September 2017 geomagnetic storm over the Europe-African longitude sector.== *J. Geophys. Res.: Space Physics*, 125 (10), e2020JA028307, https://doi.org/10.1029/2020JA028307, ==2020==]

**A list of changes and a rebuttal against each point, which has been raised by Topical Editor in Comments to the Authors.**

**Topical Editor Decision: Publish subject to revisions (further review by editor and referees)** (18 Mar 2021) by Dalia Buresova
Comments to the Author:
Dear author, dear co-authors,
**(1) Comment to the Authors:**
The manuscript contains the results of interesting measurements and provides information about the effects of an ionospheric storm on the propagation of radio waves over a certain locality, which is useful information for the scientific community. Unfortunately, in its current form, the article still gives the impression of inconsistency: a large amount of diverse information without proper interpretation and discussion.

**Authors' reply:**
According to D. Burešová "… the article still gives the impression of inconsistency…". This is a seeming contradiction. In fact, the authors have performed sufficiently detailed analysis of the solar wind parameters, the state of space weather, the parameters of the geospace storm and its components, i.e. magnetic and ionospheric storms. Further, the effect of the ionospheric storm on the Doppler spectra and signal amplitudes for 11 radio wave propagation paths is described in detail. Such an analysis is fully consistent with the goal we have address in this study, and which, as it seems to us, has been fully achieved.

**(2) Comment to the Authors:**
Please, clarify for yourself what was the main goal of your research, the results of which you wish to publish. In your responses you stated that "The subject of this study is the influence of the ionospheric storm on the characteristics of HF radio waves propagating over the area of the People's Republic of China". If this is indeed the case, please focus particularly on this goal. Then it is not necessary to analyze in detail the individual solar and geomagnetic parameters of the storm, because in the article you do not analyze how this or that specific parameter affected the observed changes in the propagation of radio waves. On the other hand, in the manuscript we are reading "This study provides general analysis of the 30 August–2 September 2019 geospace storm, the analysis of disturbances in the geomagnetic field and in the ionosphere, as well as the influence of the ionospheric storm on

the characteristics of HF radio waves over the People's Republic of China." If you are analysing both the storm occurrence, its course and the effects on HF radio wave propagation, then you should give information on storm characteristics, discuss the physical mechanism of storm behaviour and also how individual features/parameters of the storm affect the propagation of radio waves.

**Authors' reply:**

Our goal is twofold: the analysis of the storms and disturbances in the characteristics of radio waves. These are two sides of the same coin. Using the perturbations in the characteristics of the radio waves observed, we judge about dynamic processes in the ionosphere.

It is well-known fact that negative ionospheric storms are accompanied by a significant decrease in the electron density, when the level of reflection shifts upward in altitude by tens or even hundreds kilometers. Furthermore, storms generated AGWs that modulate the electron density and the Doppler shift of frequency. Both of these are manifested in the signal amplitude (see Table 2 added to the manuscript).

In reality, we could not have analyzed such questions as "how this or that specific parameter affected the observed changes in the propagation of radio waves" and "how individual features/parameters of the storm affect the propagation of radio waves".

The main reason for this is the paucity of authors' observations, only the Doppler data collected in the bottom-side F region over just a spot on the globe, and only during nighttime.

To give answers to such questions as "how individual features/parameters of the storm affect the propagation of radio waves", one must have the data on the specific processes that have really acted in the vast expanse between the solar wind and just a spot on the globe over China.

Of course, the authors could have re-written a general scheme for the processes acting during ionospheric storms. This scheme is well-known and is taught to students (e.g., Section 11.16 in the textbook by Robert W. Schunk and Andrew F. Nagy, Ionospheres Physics, Plasma Physics, and Chemistry, Second Edition, CUP, 2009).

However, the presentation of common knowledge from a textbook in a scientific paper is not adding anything valuable except for just academic speculation.

Now regarding the title of the paper.

At first, the title of the paper was "*Dynamic processes in the magnetic field and in the ionosphere during the 30 August–2 September, 2019 geospace storm*" Then the Topical Editor suggested to incorporate into the title something regarding radio waves, which resulted in the second title "*Influence of 31 August – 1 September, 2019 ionospheric storm on HF radio wave propagation*". After this change, we must "clarify for yourself what was the main goal of your research, the results of which you wish to publish". Now, we propose to combine both these titles as follows:

" *Dynamic processes in the magnetic field and in the ionosphere during the 30 August–2 September, 2019 geospace storm: Influence on HF radio wave characteristics*"

**(3) Comment to the Authors:**

Recently, your Conclusions are focused mostly on the storm classification, parameters and changes in the ionospheric parameters. There is no convincing summary/findings on the effects on the radio wave propagation. The only finding on HF wave propagation you gave here is that "In the course of the ionospheric storm, the altitude of reflection of radio waves could exhibit sharp increases from ~150 km to ~300–310 km". However, in your response you mentioned that this could be due to ionospheric transition from the day to the night time. Yes, it seems to be a most possible reason.

To see the effects of the storm on the reflexion height, you should compare measured values with the mean values of the reflexion heights (or the values for quiet days) for the same time period.

**Authors' reply:**

We have described the main characteristics of storms: their duration, moments of beginning and end, their strength (energy), individual features.

We have added section 7.3.4 and a Table 2 to the Discussion section, where the variations in the characteristics of radio waves are described in detail.

We have emphasized that at night the influences of the ionospheric storm on the characteristics of HF radio waves are exhibited more strongly, since the radio waves are reflected in the more disturbed F-region of the ionosphere.

To identify storm effects, we compared measurements for disturbed days (August 31 and September 1, 2019) with measurements for a quiet day (August 30, September 2 and 3, 2019).

**(4) Comment to the Authors:**

In the Conclusions you also stated that "The geospace storm acted to notably disturb the ionospheric E region, as well as sporadic Es layer." The Es layer is sporadic layer. It could appear during the storm, but this is not a rule. It could be formatted under quiet conditions during different seasons or time of the day and could have different duration. What you have observed, are changes in the critical frequency foEs (again, comparison with mean values or quiet time values would be useful).

**Authors' reply:**

Regarding the Es layer. We can only assert that during the period of the ionospheric storm, the foEs frequency increased from 3 MHz to 6–7 MHz. In the abstract and conclusions, we added that this is **possibly** related to the storm.

**(5) Comment to the Authors:**

Abstract and Introduction: You have newly introduced a statement "The concept that geospace storms are comprised of synergistically coupled magnetic storms, ionospheric storms, atmospheric storms, and storms in the electric field originating in the magnetosphere, the ionosphere and the atmosphere (i.e., electrical storms) was validated a few decades ago." This is correct. Nevertheless the another sentence in the abstract says that "The geospace storm was accompanied by a moderate to strong negative ionospheric storm", what evocates that ionospheric disturbances are out of the "synergistically coupled" system. Please, make a correction, for example "…during this geospace storm significant negative ionospheric effects were observed above China…" or so (the same for Conclusions)

**Authors' reply**

Quite an embarrassing error in our English. Indeed, the word "accompanied" does not belong in this phrase. Thank you very much.

We have removed the phrase ""The geospace storm was accompanied by…""everywhere in the manuscript.

Some remarks with regards to previous comments:

**(6) Remark (1) to the Authors:**

Remark with regards to previous comment **(8)**:

(8) Comment to the Authors:

"The main feature of this geospace storm is its duration, of up to four days."(between lines 70-75). The duration of storms could be quite different depending on driver (from several hours up to 8-10 days).
Authors' reply:
We have made a more accurate as "One of the interesting features of this geospace storm is its duration, of up to four days."
Dear authors, I could not agree with the change you made in the sentence above ("interesting" instead of "important"). The duration of the storm in this case is not exceptional or interesting, so there is enough to state that the storm lasted four days.
The same is true for the comment (9)

**Authors' reply**

We have omitted all phrases containing the duration of the storm as their feature.

**(7) Remark (2) to the Authors:**

Remark with regards to previous comment **(9)**:
(9) Comment to the Authors:
"Thus, this magnetic storm had the longest duration observed over the last few years,…" (part 135-140).
This statements is also doubtful.
Authors' reply:
We have made a more accurate assertion: "Thus, this magnetic storm was seen to be of quite a long duration over the last few years, ".

**Authors' reply**

We have omitted all phrases containing the duration of the storm as their feature.

**(8) Remark (3) to the Authors:**

Again, as for its duration the storm you have analysed is not exceptional when you consider that it is a storm of CIR-origin. Usually these storms have longer (more than four days) duration when comparing with those CME-related storms and the total energy input into Earth's upper atmosphere could also be larger. Going back into last few years we had several storms with similar or longer duration both CME- or CIR-related (e.g. June and March 2015; March, May, July, September, November 2017; April, August 2018)

**Authors' reply**

We have omitted all phrases containing the duration of the storm as their feature.

**(9) Remark (4) to the Authors:**

Remark with regards to previous comment:

Please, try to modify the article once again in the light of the reviewers' recommendations. The work contains interesting experimental results. It would be good to organize the results logically, put them into context and discuss them appropriately. Then the article will be understandable and beneficial for the reader.

**Authors' reply:**

The authors have tried to give answers to all Topical Editor's comments.

Kindest regards

Yours cordially
Dalia Buresova

Best regards,
Authors.